# Finite-Time Analysis of Temporal Difference Learning with Experience Replay

**Han-Dong Lim**                                                  *limaries30@kaist.ac.kr*
*Department of Electrical Engineering*
*Korea Advanced Institute of Science and Technology*

**Donghwan Lee**                                                  *donghwan@kaist.ac.kr*
*Department of Electrical Engineering*
*Korea Advanced Institute of Science and Technology*

**Reviewed on OpenReview:** *https://openreview.net/forum?id=A5ulGfDBON*

## Abstract

Temporal-difference (TD) learning is widely regarded as one of the most popular algorithms in reinforcement learning (RL). Despite its widespread use, it has only been recently that researchers have begun to actively study its finite time behavior, including the finite time bound on the mean squared error and sample complexity. On the empirical side, experience replay has been a key ingredient in the success of deep RL algorithms, but its theoretical effects on RL have yet to be fully understood. In this paper, we present a simple decomposition of the Markovian noise terms and provide finite-time error bounds for tabular on-policy TD-learning with experience replay. Specifically, under the Markovian observation model, we demonstrate that for both the averaged iterate and final iterate cases, the error term induced by a constant step-size can be effectively controlled by the size of the replay buffer and the mini-batch sampled from the experience replay buffer.

## 1 Introduction

The pioneering Deep Q-network (DQN) (Mnih et al., 2015) has demonstrated the vast potential of reinforcement learning (RL) algorithms, having achieved human-level performances in numerous Atari games (Bellemare et al., 2013). Such successes have fueled extensive research efforts in the development of RL algorithms, e.g., Sewak (2019); Mnih et al. (2016) to name just a few. Beyond video games, RL has showcased notable performance in various fields, including robotics (Singh et al., 2020) and finance (Liu et al., 2020).

On the other hand, temporal-difference (TD) learning (Sutton, 1988) is considered one of the most fundamental and well-known RL algorithms. Its objective is to learn the value function, which represents the expected sum of discounted rewards following a particular policy. While asymptotic convergence of TD-learning (Jaakkola et al., 1993; Bertsekas & Tsitsiklis, 1996) has been extensively studied and is now well-understood, such asymptotic analysis cannot measure how efficiently the estimation progresses towards a solution. Recently, the convergence rate of TD-learning has gained much attention and has been actively investigated (Bhandari et al., 2018; Srikant & Ying, 2019; Lee & Kim, 2022). These studies aim to understand the efficiency of the estimation process, and provide theoretical guarantees on the rate of convergence.

First appearing in Lin (1992), experience replay memory can be viewed as a first-in-first-out queue and is one of the principal pillars of DQN (Mnih et al., 2015). Learning through uniformly random samplings from the experience replay memory, the strategy is known to reduce correlations among experience samplings, and improve the efficiency of the learning. Despite its empirical successes (Mnih et al., 2015; Fedus et al., 2020; Hong et al., 2022), the theoretical side of the experience replay memory techniques remains largely an open yet challenging question. Only recently, Di-Castro et al. (2022; 2021) studied asymptotic convergence of actor-critic algorithm (Konda & Tsitsiklis, 1999) with experience replay memory. To the authors' knowledge, its non-asymptotic analysis has not been thoroughly investigated in the context of TD-learning so far.

The aim of this paper is to investigate the impact of experience replay memory on TD-learning (Sutton, 1988), with the goal of shedding light on the question at hand. Our primary theoretical contribution is the derivation of the convergence rate of tabular on-policy TD-learning with experience replay memory under Markovian observation models. This analysis reveals connections between the sizes of the mini-batch and experience replay memory, and the convergence rate. Specifically, we show that the error term, resulting from correlations among the samples from the Markovian observation models, can be effectively controlled by the sizes of the mini-batch and experience replay memory, for both the averaged and final iterate cases. We expect that our findings can provide further insights into the use of experience replay memory in RL algorithms.

Lastly, although our analysis only considers the standard TD-learning case, our presented arguments can be extended to more general scenarios, such as TD-learning with linear function approximation (Srikant & Ying, 2019), standard Q-learning (Watkins & Dayan, 1992), periodic Q-learning (Lee & He, 2019).

### 1.1 Related works

**Experience replay.** We begin by providing an overview of the existing theoretical results on experience replay. The recent work by Nagaraj et al. (2020) leverages the experience replay memory to address the least-squares problem under the Gaussian auto-regressive model. However, there are several notable differences between their approach and the proposed TD-learning with experience replay memory:

1. The approach in Nagaraj et al. (2020) assumes i.i.d. Gaussian noises, whereas the proposed TD-learning with experience replay memory covers Markovian and specific non-Gaussian noises.

2. The overall algorithmic structures are significantly different. Nagaraj et al. (2020) considered an offline learning problem, while the proposed TD-learning framework is an online learning approach.

3. When operating the experience replay buffer, they maintain a sufficiently large gap between the separate samples inside the buffer to ensure the samples are almost identically and independently distributed.

4. The approach by Nagaraj et al. (2020) uses an inner loop to iterate over the samples in the buffer, whereas the proposed framework considers mini-batch style updates, which is widely used in practice.

Kowshik et al. (2021) presented an algorithm that employs the reverse experience memory approach proposed in Rotinov (2019) to tackle linear system identification problems. This algorithm guarantees non-asymptotic convergence, and the reverse experience replay technique (Rotinov, 2019) involves using transitions in reverse order, rather than uniformly and randomly, without introducing any stochasticity, which distinguishes it from the original experience replay method introduced in Mnih et al. (2015). Kowshik et al. (2021) assumes an auto-regressive model with i.i.d. noise. Following the spirit of Nagaraj et al. (2020), Kowshik et al. (2021) also maintains a gap between separate samples to enforce independence between the samples.

Using the super-martingale structure of the reverse experience replay memory, Agarwal et al. (2021) derives the sample complexity of Q-learning, which also maintains a gap between samples. Moreover, the works, Kowshik et al. (2021) and Agarwal et al. (2021) both exploit full samples from each experience replay memory rather than applying uniformly sampled mini-batch to maintain the reverse order property.

Lazic et al. (2021) presented a regret bound analysis for regularized policy iteration with a replay buffer in the context of averaged reward Markov decision processes (MDPs). The authors assume that an accurate estimate of the action-value function is available, which is obtained via Monte Carlo methods (Singh & Sutton, 1996), as opposed to the TD-learning algorithm (Sutton, 1988). The use of experience replay memory is modified from Mnih et al. (2015). One approach suggested in Lazic et al. (2021) involves storing all data from each phase, with a limit on the size of the replay buffer. When the buffer size exceeds the limit, a subset of the data is eliminated uniformly at random.

Fan et al. (2020) established a non-asymptotic convergence of fitted Q-learning (Ernst et al., 2005), which assumes i.i.d. sampling of transitions from a fixed distribution. This assumption is stronger compared to the Markovian observation models used with the time-varying replay buffer. The Markovian observation model is a more natural and realistic scenario than the i.i.d. modeling, which improves practicality of the

analysis. Furthermore, The replay buffer technique is different from the sampling i.i.d. transitions because in the mini-batch approach, the samples are from a single episode trajectory in a moving window, which are highly correlated, and the correlation increases proportionally to the size of the window. Lastly, Fan et al. (2020) imposed a condition that the fixed distribution satisfies a particular condition such that it is similar to the true underlying distribution over the state-action space, which may not hold in our setting. Moreover, the fitted Q-iteration is similar to stochastic gradient descent algorithm in supervised-learning problem, whereas TD-learning is an online learning algorithm which does not use any (stochastic) gradient of an objective function.

A closely related approach to our work is presented in Di-Castro et al. (2022), which establishes the asymptotic convergence of the actor-critic algorithm (Konda & Tsitsiklis, 1999) using a mini-batch that is uniformly and randomly sampled from the replay buffer. The proof relies on treating the replay buffer and mini-batch indices as random variables, which together form an irreducible and aperiodic Markov chain. Following the O.D.E. approach outlined in Borkar & Meyn (2000) and the natural actor-critic algorithm described in Bhatnagar et al. (2009), Di-Castro et al. (2022) establishes the convergence of the actor-critic algorithm using the replay buffer. While Di-Castro et al. (2022) demonstrates a decrease in auto-correlation and covariance among samples in the experience replay, their proof of asymptotic convergence does not explicitly address the impact of these factors on the convergence behavior. However, the study provides important insights into the convergence properties of actor-critic algorithms using the replay buffer and highlights the potential benefits of using mini-batches to improve the convergence rate.

**Non-asymptotic analysis of TD-learning** We highlight several recent studies on finite time behavior of TD-learning (Lee & Kim, 2022; Bhandari et al., 2018; Chen et al., 2020; Srikant & Ying, 2019; Dalal et al., 2018; Hu & Syed, 2019). Under an i.i.d. observation model and linear function approximation setting, Dalal et al. (2018) derived $O\left(\frac{1}{k^\sigma}\right)$ bound on the mean squared error with diminishing step-size $\frac{1}{(k+1)^\sigma}$ where $\sigma \in (0,1)$ and $k$ is the number of iterations. Under an i.i.d. observation model and tabular setup, from the discrete-time stochastic linear system perspective, Lee & Kim (2022) provided a geometric convergence rate with constant error at the order of $O(\alpha)$ of TD-learning for both averaged iterate and final iterate using constant step-size $\alpha$. Bhandari et al. (2018) provided a convergence rate of TD-learning with linear function approximation under the Markovian noise following the spirit of convex optimization literature (Nemirovski et al., 2009). With the help of Moreau envelope (Parikh et al., 2014), Chen et al. (2020) derived a convergence rate of TD-learning with respect to an arbitrary norm. Mainly based on the Lyapunov approach (Khalil, 2015) for continuous time O.D.E. counterpart of TD-learning, Srikant & Ying (2019) derived a finite time bound on the mean squared error of TD-learning under Markovian noise with linear function approximation.

In contrast to the existing studies that have focused on the finite time behavior of TD-learning (Lee & Kim, 2022; Bhandari et al., 2018; Chen et al., 2020; Srikant & Ying, 2019; Dalal et al., 2018; Hu & Syed, 2019), our work investigates the finite time behavior of TD-learning with experience replay, which has not been thoroughly studied to date. Specifically, we demonstrate that the use of experience replay can be an effective means of reducing the constant error term that arises from employing a constant step-size. Our findings may provide valuable insights into the effectiveness of experience replay in RL, shedding new light on the benefits of this widely-used technique. Further research in this area could yield important research topics, with implications for the development of more efficient and effective reinforcement learning algorithms.

## 2 Preliminaries

### 2.1 Markov chain

In this section, basic concepts of Markov chain are briefly introduced. To begin with, the so-called total variation distance defines the distance between two probability measures as follows.

**Definition 2.1** (Total variation distance (Levin & Peres, 2017))**.** *The total variation distance between two probability distributions, $\mu_1$ and $\mu_2$, on $\mathcal{S}$ is given by*

$$d_{\mathrm{TV}}(\mu_1, \mu_2) = \sup_{A \subseteq \mathcal{S}} |\mu_1(A) - \mu_2(A)|.$$

Let us consider a Markov chain with the set of states $\mathcal{S} := \{1, 2, \ldots, |\mathcal{S}|\}$ and the state transition probability $\mathcal{P}$. For instance, a state $s \in \mathcal{S}$ transits to the next state $s'$ with probability $\mathcal{P}(s, s')$. A stationary distribution of the Markov chain is defined as a distribution $\mu \in \mathbb{R}^{|\mathcal{S}|}$ on $\mathcal{S}$ such that $\mu^\top P = \mu^\top$ where $P \in \mathbb{R}^{|\mathcal{S}| \times |\mathcal{S}|}$ is the transition matrix of the Markov chain, i.e., $[P]_{ij} = \mathcal{P}(i, j)$ for $i, j \in \mathcal{S}$.

Let $\{S_k\}_{k \geq 0}$ be a trajectory of a Markov chain. Then, an irreducible and aperiodic Markov chain is known to admit a unique stationary distribution $\mu$ such that the total variation distance between the stationary distribution and the current state distribution decreases exponentially (Levin & Peres, 2017) as follows:

$$\sup_{s \in \mathcal{S}} d_{\mathrm{TV}}(\mathbb{P}[S_k = \cdot \mid S_0 = s], \mu) \leq m\rho^k$$

for some $\rho \in (0, 1)$ and positive $m \in \mathbb{R}$. The mixing time of a Markov chain is defined as

$$t^{\mathrm{mix}}(\epsilon) := \min \left\{ k \in \mathbb{Z}_+ : \sup_{s \in \mathcal{S}} d_{\mathrm{TV}}(\mathbb{P}[S_k = \cdot \mid S_0 = s], \mu) \leq \epsilon \right\} \tag{1}$$

for any $\epsilon \in \mathbb{R}_+$. Throughout the paper, we will use $t^{\mathrm{mix}}$ to denote $t^{\mathrm{mix}}\left(\frac{1}{4}\right)$ for simplicity.

## 2.2 Markov decision process

A Markov decision process is described by the tuple $(\mathcal{S}, \mathcal{A}, \gamma, \mathcal{P}, r)$, where $\mathcal{S} := \{1, 2, \ldots, |\mathcal{S}|\}$ is the set of states, $\mathcal{A} := \{1, 2, \ldots, |\mathcal{A}|\}$ is the set of actions, $\gamma \in (0, 1)$ is the discount factor, $r : \mathcal{S} \times \mathcal{A} \times \mathcal{S} \to \mathbb{R}$ is the reward function, and $\mathcal{P} : \mathcal{S} \times \mathcal{A} \times \mathcal{S} \to [0, 1]$ is the state transition probability, i.e., $\mathcal{P}(s, a, s')$ means the probability of the next state $s' \in \mathcal{S}$ when taking action $a \in \mathcal{A}$ at the current state $s \in \mathcal{S}$. For example, at state $s_k \in \mathcal{S}$ at time $k$, if an agent selects an action $a_k$, then the state transits to the next state $s_{k+1}$ with probability $\mathcal{P}(s_k, a_k, s_{k+1})$, and incurs the reward $r(s_k, a_k, s_{k+1})$, where the reward generated by the action at time $k$, $r(s_k, a_k, s_{k+1})$, will be denoted by $r_{k+1} := r(s_k, a_k, s_{k+1})$. In this paper, we adopt the standard assumption on the boundedness of the reward function.

**Assumption 2.2.** *There exists a positive $R_{\max} \in \mathbb{R}$ such that $|r(s, a, s')| \leq R_{\max}$ for all $s, a, s' \in \mathcal{S} \times \mathcal{A} \times \mathcal{S}$.*

Let us consider a Markov decision process with the policy, $\pi : \mathcal{S} \times \mathcal{A} \to [0, 1]$. The corresponding state trajectory, $\{S_k\}_{k \geq 0}$, is a Markov chain induced by the policy $\pi$, and the state transition probability is given by $\mathcal{P}^\pi : \mathcal{S} \times \mathcal{S} \to [0, 1]$, i.e., $\mathcal{P}^\pi(s, s') := \sum_{a \in \mathcal{A}} \mathcal{P}(s, a, s')\pi(a \mid s)$ for $s, s' \in \mathcal{S}$. Throughout the paper, we assume that the induced Markov chain with transition kernel $\mathcal{P}^\pi$ is irreducible and aperiodic so that it admits a unique stationary distribution denoted by $\mu_{S_\infty}^\pi$, and satisfies the exponential convergence property

$$\sup_{s \in \mathcal{S}} d_{\mathrm{TV}}(\mathbb{P}[S_k = \cdot \mid S_0 = s], \mu_{S_\infty}^\pi) \leq m_1 \rho_1^k, \quad k \geq 0, \tag{2}$$

for some positive $m_1 \in \mathbb{R}$ and $\rho_1 \in (0, 1)$. Let $(S_k, S_{k+1}) \in \mathcal{O}$ be a tuple of states at time step $k$ and its next state $S_{k+1} \sim \mathcal{P}^\pi(S_k, \cdot)$, which will be frequently used in this paper to analyze TD-learning, and $\mathcal{O}$ denotes the realizable set of tuples consisting of a state and its next state, i.e., for $(x, y) \in \mathcal{S} \times \mathcal{S}$, we have $(x, y) \in \mathcal{O}$ if and only if $\mathcal{P}^\pi(x, y) > 0$. Then, the tuple forms another induced Markov chain. The transition probability of the induced Markov chain $\{(S_k, S_{k+1})\}_{k \geq 0}$ is $\mathbb{P}[S_{k+2}, S_{k+1} \mid S_{k+1}, S_k] = \mathbb{P}[S_{k+2} \mid S_{k+1}]$, which follows from the Markov property. The next lemma states that the Markov chain $\{(S_k, S_{k+1})\}_{k \geq 0}$ is also irreducible and aperiodic provided that $\{S_k\}_{k \geq 0}$ is irreducible and aperiodic.

**Lemma 2.3.** *If $\{S_k\}_{k \geq 0}$ is an irreducible and aperiodic Markov chain, then so is $\{(S_k, S_{k+1})\}_{k \geq 0}$.*

The proof is in Appendix Section A.3. Now, let us denote $\mu_{S_\infty, S_\infty'}^\pi$ as the stationary distribution of the Markov chain $\{(S_k, S_{k+1})\}_{k \geq 0}$, which satisfies the relation between $\mu_{S_\infty}^\pi$ and $\mu_{S_\infty, S_\infty'}^\pi$:

$$\sum_{a \in \mathcal{A}} \pi(a \mid s)\mathcal{P}(s, a, s')\mu_{S_\infty}^\pi(s) = \mathcal{P}^\pi(s, s')\mu_{S_\infty}^\pi(s) = \mu_{S_\infty, S_\infty'}^\pi(s, s'). \tag{3}$$

Then, from Lemma 2.3, we have $\sup_{(s,s') \in \mathcal{S} \times \mathcal{S}} d_{\mathrm{TV}}\left(\mathbb{P}[(S_k, S_{k+1}) = \cdot \mid (S_0, S_1) = (s, s')], \mu_{S_\infty, S_\infty'}^\pi\right) \leq m_2 \rho_2^k$, for some positive $m_2 \in \mathbb{R}$ and $\rho_2 \in (0, 1)$. Similar to the original Markov chain, the mixing time can be

defined for this new Markov chain. Throughout this paper, we adopt the notations, $t_1^{\text{mix}}$ and $t_2^{\text{mix}}$, to denote the mixing time of the Markov chain $\{S_k\}_{k \geq 0}$ and the mixing time of $\{(S_k, S_{k+1})\}_{k \geq 0}$, respectively. For simplicity of the notation, we will denote $\tau^{\text{mix}} := \max\{t_1^{\text{mix}}, t_2^{\text{mix}}\}$.

### 2.3 Temporal difference learning

To begin with, several matrix notations are introduced. Let us define

$$
D^\pi := \begin{bmatrix} \mu_{S_\infty}^\pi(1) & & & \\ & \mu_{S_\infty}^\pi(2) & & \\ & & \ddots & \\ & & & \mu_{S_\infty}^\pi(|\mathcal{S}|) \end{bmatrix} \in \mathbb{R}^{|\mathcal{S}| \times |\mathcal{S}|}, \quad R^\pi = \begin{bmatrix} \mathbb{E}[r(s,a,s')|s=1] \\ \mathbb{E}[r(s,a,s')|s=2] \\ \vdots \\ \mathbb{E}[r(s,a,s')|s=|\mathcal{S}|] \end{bmatrix} \in \mathbb{R}^{|\mathcal{S}|},
$$

where $R^\pi$ defined above is a vector of expected rewards when action is taken under $\pi$. From Assumption 2.2, one can readily prove that $R^\pi$ is bounded as well.

**Lemma 2.4.** *We have* $\|R^\pi\|_\infty \leq R_{\max}$.

Moreover, in this paper, $P^\pi \in \mathbb{R}^{|\mathcal{S}| \times |\mathcal{S}|}$ denotes the transition matrix under the policy $\pi$, i.e., $[P^\pi]_{ij} = \mathcal{P}^\pi(i,j)$ for $i, j \in \mathcal{S}$, where $[P^\pi]_{ij}$ denotes the element in the $i$-th column and $j$-th row. The minimum probability in the stationary state distribution is denoted by $\mu_{\min}^\pi := \min_{s \in \mathcal{S}} \mu_{S_\infty}^\pi(s)$.

Proposed in Sutton (1988), TD-learning aims to estimate the value function $V^\pi(s) := \sum_{k=0}^\infty \mathbb{E}\left[\gamma^k r_k \mid S_0 = s, \pi\right]$, $s \in \mathcal{S}$ through the following stochastic recursion for $k \geq 0$:

$$
V_{k+1} = V_k + \alpha \delta(O_k, V_k), \tag{4}
$$

where $\alpha \in (0, 1)$ is a constant step-size, $O_k := (s_k, r_{k+1}, s_{k+1})$ , and $\delta$ is called the TD-error defined as

$$
\delta(O_k, V_k) := e_{s_k} r_k + \gamma e_{s_k} e_{s_{k+1}}^\top V_k - e_{s_k} e_{s_k}^\top V_k, \tag{5}
$$

where $e_{s_k}$ is the basis vector in $\mathbb{R}^{|\mathcal{S}|}$ whose $s_k$-th element is one and all others are zero.

## 3 Main results

### 3.1 TD-learning with experience replay

In this subsection, we introduce the proposed TD-learning method employing the original experience replay memory technique from Mnih et al. (2015) with no significant modifications. Experience replay memory, commonly referred to as the replay buffer, is a first-in-first-out (FIFO) queue that facilitates the adoption of mini-batch techniques in machine learning for online learning scenarios. Specifically, the replay buffer stores the state-action-reward transitions in a FIFO manner, serving as the training set. At each step, a mini-batch is uniformly sampled from the replay buffer and utilized to update the learning parameter $V_k$. This approach presents dual advantages. Firstly, it enables the application of the batch update scheme, thereby reducing variance and accelerating learning. Secondly, through uniform sampling, it may reduce correlations among different samples in the Markovian observation models and consequently reduce extra biases in the value estimation. Despite its empirical benefits, theoretical studies on replay buffer have been limited.

In this paper, we employ the notation $B_k^\pi$ and $M_k^\pi$ to represent the replay buffer and mini-batch, respectively, at time step $k$. They are formally defined as follows:

$$
M_k^\pi := \{O_1^k, O_2^k \ldots, O_L^k\}, \quad B_k^\pi := \{O_{k-N+1}, O_{k-N+2} \ldots, O_k\},
$$

where the replay buffer's size is $N$, the mini-batch's size is $L$, and $O_i^k, 1 \leq i \leq L$ stands for the $i$-th sample of the mini-batch $M_k^\pi$. For the first $N$ steps, only the observations, $O_{-N}$ from $O_{-1}$ , is collected without updating the $V_k$. That is, at time-step $k$, $O_k$ is observed and $V_k$ is updated using a mini-batch $M_k^\pi$ sampled from batch $B_k^\pi$. The overall scheme is depicted in Figure 1.

For each time step $k \geq 0$, the agent selects an action $a_k$ following the target policy $\pi$, and observes next state $s_{k+1}$. The oldest sample $(s_{k-N}, a_{k-N}, s_{k-N+1})$ is dropped from the replay buffer $B_{k-1}^\pi$, and the new sample $(s_k, a_k, s_{k+1})$ is added to the replay buffer, which becomes $B_k^\pi$.

Next, a mini-batch of size $L$, $M_k^\pi$, is sampled uniformly from the replay buffer $B_k^\pi$ at time step $k$, and the $k+1$-th iterate of TD-learning is updated via

$$V_{k+1} = V_k + \alpha_k \frac{1}{|M_k^\pi|} \sum_{(s,r,s')\in M_k^\pi} (e_s)(r + \gamma e_{s'}^\top V_k - e_s^\top V_k), \qquad (6)$$

which is a batch update version of the standard TD-learning. Note that when $|B_k^\pi| = |M_k^\pi| = 1$ for all $k \geq 0$, the update in (6) matches that of the standard TD-learning in (4). In this paper, we consider a constant step-size $\alpha \in (0,1)$, and the Markovian observation model, which means that transition samples are obtained from a single trajectory of the underlying Markov decision process.

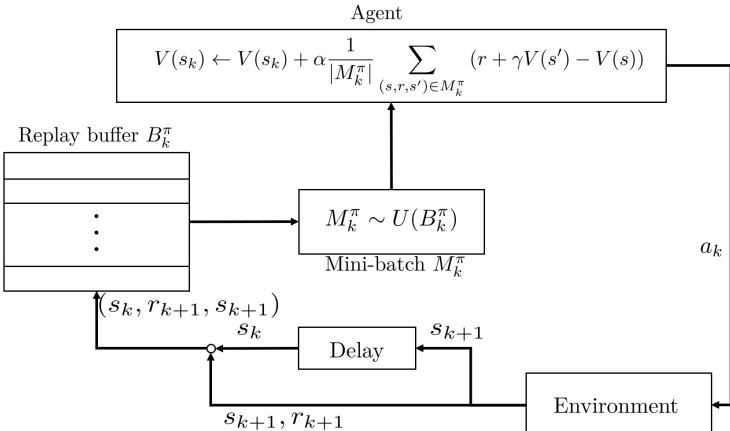

Figure 1: Diagram of TD-learning using experience replay

---

**Algorithm 1** TD-learning with replay buffer

---

1: Initialize $V_0 \in \mathbb{R}^{|\mathcal{S}|}$ such that $||V_0||_\infty \leq \frac{R_{\max}}{1-\gamma}$.
2: Collect $N$ samples : $B_{-1} := \{O_{-N}, O_{-N+1}, \ldots, O_{-1}\}$.
3: **for** $k \leq T$ **do**
4:     Select action $a_k \sim \pi(\cdot|s_k)$.
5:     Observe $s_{k+1} \sim \mathcal{P}(\cdot|s_k, a_k)$ and recieve reward $r_{k+1} := r(s_k, a_k, s_{k+1})$.
6:     Update replay buffer : $B_k^\pi \leftarrow B_{k-1}^\pi \setminus \{(s_{k-N}, r_{k-N+1}, s_{k-N+1})\} \cup \{(s_k, r_{k+1}, s_{k+1})\}$.
7:     Uniformly sample $M_k^\pi \sim B_k^\pi$.
8:     Update $V_{k+1} = V_k + \alpha_k \frac{1}{|M_k^\pi|} \sum_{(s,r,s')\in M_k^\pi}(e_s)(r + \gamma e_{s'}^\top V_k - e_s^\top V_k)$.
9: **end for**

---

To proceed with our analysis, we should establish the boundedness of the iterate resulting from the update in (6), assuming that $||V_0||_\infty \leq \frac{R_{\max}}{1-\gamma}$ and that $\alpha_k \in (0,1)$. This assumption is crucial to our main developments, and thus requires rigorous proof.

**Lemma 3.1.** *Under the recursion in (6), $V_k$ remains bounded :* $||V_k||_\infty \leq \frac{R_{\max}}{1-\gamma}$.

The proof is given in Appendix Section A.4.

### 3.2 Analysis framework

In the previous subsection, the algorithm underlying our analysis was introduced. In this subsection, we provide the preliminary frameworks for our main analysis. Specifically, we analyze TD-learning based partially on the linear dynamical system viewpoint, as presented in the recent work by Lee & Kim (2022). In particular, using the Bellman equation (Bertsekas & Tsitsiklis, 1996), $D^\pi V^\pi = \gamma D^\pi P^\pi V^\pi + D^\pi R^\pi$, we can express the TD-learning update (6) as follows:

$$V_{k+1} - V^\pi := A(V_k - V^\pi) + \alpha w(M_k^\pi, V_k), \qquad (7)$$

where

$$A := I - \alpha D^\pi + \alpha \gamma D^\pi P^\pi \tag{8}$$

is the system matrix, and

$$w(M_k^\pi, V_k) := \frac{1}{|M_k^\pi|} \sum_{(s,r,s') \in M_k^\pi} (e_s)(r + \gamma e_{s'}^\top V_k - e_s^\top V_k) - D^\pi(R^\pi + \gamma P^\pi V_k - V_k) \tag{9}$$

is the noise, which is the difference between the empirical mean of the TD-error via samples in mini-batch and the expected TD-error with respect to the stationary distribution $\mu_{S_\infty}^\pi$.

Some useful properties of the system matrix $A$ (boundedness of $A$, and related Lyapunov theory (Chen, 1984)) are introduced in the next lemma, which play central roles in establishing the convergence rate.

**Lemma 3.2** (Properties of matrix $A$ (Lee & Kim, 2022)).

1) $||A||_\infty \leq 1 - \alpha(1 - \gamma)\mu_{\min}^\pi$ holds.

2) There exists a positive definite matrix $M \succ 0$ and $||M||_2 \leq \frac{2|\mathcal{S}|}{\alpha(1-\gamma)\mu_{\min}^\pi}$ such that

$$A^\top M A - M = -I. \tag{10}$$

For completeness of presentation, the proof is given in Appendix A.5. To proceed, let us define the empirical distributions for $i, j \in \mathcal{S}$,

$$\mu_S^{B_k^\pi}(i) := \frac{\sum_{(s,r,s') \in B_k^\pi} 1\{s = i\}}{|B_k^\pi|}, \quad \mu_{S,S'}^{B_k^\pi}(i,j) := \frac{\sum_{(s,r,s') \in B_k^\pi} 1\{(s,s') = (i,j)\}}{|B_k^\pi|},$$

which denote the empirical distributions of the state $s$ and the tuple $(s, s')$ in the replay buffer $B_k^\pi$, respectively. Moreover, for an event $A$, the notation $1\{A\}$ denotes an indicator function that returns one if the event $A$ occurs and otherwise zero. Furthermore, let us introduce the following matrix notations for $i, j \in \mathcal{S}$:

$$[D^{B_k^\pi}]_{ij} := \begin{cases} \mu_S^{B_k^\pi}(i) & \text{if} \quad i = j \\ 0 & \text{otherwise} \end{cases}, \quad [P^{B_k^\pi}]_{ij} := \begin{cases} \frac{\sum_{(s,r,s') \in B_k^\pi} 1\{(s,s') = (i,j)\}}{\sum_{(s,r,s') \in B_k^\pi} 1\{s = i\}} & \text{if} \quad |B^\pi(i)| \geq 1 \\ 0 & \text{otherwise} \end{cases},$$

which define the empirical distribution of the state in the replay buffer and $B_k^\pi(s) := \{(\tilde{s}, \tilde{r}, \tilde{s}') \in B_k^\pi : \tilde{s} = s\} \subseteq B^\pi$. With above definitions, we can readily verify the relation $[D^{B_k^\pi} P^{B_k^\pi}]_{ij} = \mu_{S,S'}^{B_k^\pi}(i,j)$ for $i, j \in \mathcal{S}$.

Likewise, let us define the empirical estimation of the expected return, which is averaged over the action, $a$, and next state, $s'$. It is calculated from the samples of the replay buffer as follows for $i \in \mathcal{S}$:

$$[R^{B_k^\pi}]_i = \begin{cases} \frac{\sum_{(s,r,s') \in B_k^\pi} 1\{s = i\}r}{\sum_{(s,r,s') \in B_k^\pi} 1\{s = i\}} & \text{if} \quad |B_k^\pi(i)| \geq 1 \\ 0 & \text{otherwise} \end{cases}.$$

## 3.3 Bounds on noise

Our aim in this subsection is to bound the first and second moment of $||w(M_k^\pi, V_k)||_2$ where $w(M_k^\pi, V_k)$ is defined in (9), which will play important role in deriving the convergence rate. For simplicity, let us further define the functions, $\Delta_k : \mathbb{R}^{|\mathcal{S}|} \to \mathbb{R}^{|\mathcal{S}|}$ and $\Delta_\pi : \mathbb{R}^{|\mathcal{S}|} \to \mathbb{R}^{|\mathcal{S}|}$, as follows:

$$\Delta_k(V) = D^{B_k^\pi} R^{B_k^\pi} + \gamma D^{B_k^\pi} P^{B_k^\pi} V - D^{B_k^\pi} V, \tag{11}$$

$$\Delta_\pi(V) = D^\pi R^\pi + \gamma D^\pi P^\pi V - D^\pi V. \tag{12}$$

The functions, $\Delta_k$ and $\Delta_\pi$, can be viewed as expected TD-errors, respectively, in terms of the distribution of the replay buffer and the stationary distribution of the Markov chain. Moreover, note that $\Delta_\pi(V^\pi) = 0$. Based on the notations introduced, the noise term, $w(M_k^\pi, V_k)$, can be decomposed into the two parts

$$w(M_k^\pi, V_k) = \frac{1}{|M_k^\pi|} \sum_{i=1}^{|M_k^\pi|} \delta(O_i^k, V_k) - \Delta_k(V_k) - (\Delta_\pi(V_k) - \Delta_k(V_k)), \tag{13}$$

where

1. $\frac{1}{|M_k^\pi|} \sum_{i=1}^{|M_k^\pi|} \delta(O_i^k, V_k) - \Delta_k(V_k)$: the difference between the empirically expected TD-error with respect to the distribution of mini-batch and empirically expected TD-error with respect to the replay buffer.

2. $\Delta_\pi(V_k) - \Delta_k(V_k)$: the difference between the expected TD-error with respect to the stationary distribution and the empirically expected TD-error with respect to the distribution of replay buffer.

The high level idea for bounding the first and second moment of $\|w(M_k^\pi, V_k)\|_2$ is summarized below.

1. $\frac{1}{|M_k^\pi|} \sum_{i=1}^{|M_k^\pi|} \delta(O_i^k, V_k) - \Delta_k(V_k)$, the error between the empirical distribution of the mini-batch and the replay buffer, can be bounded by Bernstein inequality (Tropp et al., 2015) because the mini-batch samples are independently sampled from the replay buffer with the uniform distribution.

2. $\Delta_\pi(V_k) - \Delta_k(V_k)$, the error between the stationary distribution and distribution of replay buffer, can be bounded using the property of irreducible and aperiodic Markov chain (Levin & Peres, 2017).

Next, we introduce several lemmas to derive a bound on the the first moment of $\|w(M_k^\pi, V_k)\|_2$, which can be bounded at the order of $O\left(\sqrt{\frac{1}{|M_k^\pi|}}\right)$. To bound $\Delta_\pi(V_k) - \Delta_k(V_k)$, we introduce the coupled process $\{\tilde{S}_k\}_{k \geq -N}$, which starts from the stationary distribution of the Markov chain with transition kernel $\mathcal{P}^\pi$, i.e., $\tilde{S}_{-N} \sim \mu_{S_\infty}^\pi$. Let $\tilde{B}_k^\pi$ be the corresponding replay buffer of such a Markov chain. We first derive the expected error bounds in terms of the coupled process $\{\tilde{S}_k\}_{k \geq -N}$, and the desired result will be obtained using the total variation distance between the distribution of $\tilde{S}_k$ and $S_k$. The detailed proof is given in Appendix A.6. Now, combining with the bound on $\frac{1}{|M_k^\pi|} \sum_{i=1}^{|M_k^\pi|} \delta(O_i^k, V_k) - \Delta_k(V_k)$, which follows from concentration inequalities, we get the following result:

**Lemma 3.3.** *For $k \geq 0$, $\mathbb{E}[\|w(M_k^\pi, V_k)\|_2]$ can be bounded as follows:*

$$\mathbb{E}[\|w(M_k^\pi, V_k)\|_2] \leq \frac{4\sqrt{|\mathcal{S}|}R_{\max}}{1-\gamma}\left(2\sqrt{\frac{2\log(2|\mathcal{S}|)}{|M_k^\pi|}} + 3|\mathcal{S}|^2|\mathcal{A}|\sqrt{\frac{\tau^{\mathrm{mix}}}{|B_k^\pi|}} + 16|\mathcal{S}|d_{\mathrm{TV}}(\mu_{S_\infty}^\pi, \mu_{S_{k-N}}^\pi)\right).$$

The detailed proof is given in Appendix Section A.7. Using similar arguments to bound the first moment of $\|w(M_k^\pi, V_k)\|_2$, we can bound the second moment, which is given in the following lemma.

**Lemma 3.4** (Second moment of $\|w(M_k^\pi, V_k)\|_2$). *Let us consider the noise term, $w(M_k, V_k)$, is defined in (9). For $k \geq 0$, the second moment of $\|w(M_k, V_k)\|_2$ is bounded as follows:*

$$\mathbb{E}\left[\|w(M_k^\pi, V_k)\|_2^2\right] \leq \frac{4|\mathcal{S}|(R_{\max}+1)^2}{(1-\gamma)^2}\left(\frac{120(\log(2|\mathcal{S}|))^2}{|M_k^\pi|} + 4|\mathcal{S}|^4|\mathcal{A}|^2\frac{\tau^{\mathrm{mix}}}{|B_k^\pi|} + 8|\mathcal{S}|d_{\mathrm{TV}}(\mu_{S_\infty}^\pi, \mu_{S_{k-N}}^\pi)\right).$$

The proof is deferred to Appendix Section A.8 for compactness of the presentation.

### 3.4 Averaged iterate convergence

In the previous subsection, we derived a bound on the noise term. Based on it, in this subsection, we analyze the convergence of the averaged iterate of TD-learning with experience replay. In the next theorem, we present the main result for the convergence rate on the average iterate.

**Theorem 3.5** (Convergence rate on average iterate of TD-learning)**.** *Suppose* $N > \tau^{mix}$*.*

*1) For $T \geq 0$, the following inequality holds:*

$$\frac{1}{T}\sum_{k=0}^{T-1}\mathbb{E}[||V_k - V^\pi||_2^2] \leq \frac{1}{T}\frac{2|\mathcal{S}|}{\alpha(1-\gamma)}\mathbb{E}[||V_0 - V^\pi||_2^2] + \underbrace{\frac{32|\mathcal{S}|^2 R_{\max}^2}{(1-\gamma)^3\mu_{\min}^\pi}\sqrt{\frac{8\log(2|\mathcal{S}|)}{L}}}_{E_1^{\text{avg}}:\ Concentration\ error\ in\ the\ first\ moment}$$

$$+ \underbrace{\frac{32|\mathcal{S}|^2 R_{\max}^2}{(1-\gamma)^3\mu_{\min}^\pi}\left(2|\mathcal{S}|^{\frac{3}{2}}|\mathcal{A}|\sqrt{\frac{\tau^{\text{mix}}}{N}} + \frac{64t_1^{\text{mix}}}{T}\right)}_{E_2^{\text{avg}}:\ Markovian\ noise\ in\ the\ first\ moment} \tag{14}$$

$$+ \underbrace{\alpha\frac{4|\mathcal{S}|^2(R_{\max}+1)^2}{(1-\gamma)^3\mu_{\min}^\pi}\left(\frac{120(\log 2(|\mathcal{S}|))^2}{L}\right)}_{E_3^{\text{avg}}:\ Concentration\ error\ in\ the\ second\ moment}$$

$$+ \underbrace{\alpha\frac{4|\mathcal{S}|^2(R_{\max}+1)^2}{(1-\gamma)^3\mu_{\min}^\pi}\left(4|\mathcal{S}|^4|\mathcal{A}|^2\frac{\tau^{\text{mix}}}{N} + \frac{32t_1^{\text{mix}}}{T}\right)}_{E_4^{\text{avg}}:\ Markovian\ noise\ in\ the\ second\ moment}.$$

*2) For $T \geq 0$, the following inequality holds:*

$$\mathbb{E}\left[\left\|\frac{1}{T}\sum_{k=0}^{T}V_k - V^\pi\right\|_2\right] \leq \frac{1}{\sqrt{T}}\sqrt{\frac{2|\mathcal{S}|}{\alpha(1-\gamma)}}||V_0 - V^\pi||_2 + \sqrt{\frac{32|\mathcal{S}|^2 R_{\max}^2}{(1-\gamma)^3\mu_{\min}^\pi}\sqrt{\frac{8\log(2|\mathcal{S}|)}{L}}}$$

$$+ \sqrt{\frac{32|\mathcal{S}|^2 R_{\max}^2}{(1-\gamma)^3\mu_{\min}^\pi}\left(2|\mathcal{S}|^{\frac{3}{2}}|\mathcal{A}|\sqrt{\frac{\tau^{\text{mix}}}{N}} + \frac{64t_1^{\text{mix}}}{T}\right)}$$

$$+ \sqrt{\alpha\frac{4|\mathcal{S}|^2(R_{\max}+1)^2}{(1-\gamma)^3\mu_{\min}^\pi}\frac{120(\log(2|\mathcal{S}|))^2}{L}}$$

$$+ \sqrt{\alpha\frac{4|\mathcal{S}|^2(R_{\max}+1)^2}{(1-\gamma)^3\mu_{\min}^\pi}\left(4|\mathcal{S}|^4|\mathcal{A}|^2\frac{\tau^{\text{mix}}}{N} + \frac{32t_1^{\text{mix}}}{T}\right)}.$$

The proof is given in Appendix A.9. Several comments can be made about the results in Theorem 3.5. The term (14) arises because $\mathbb{E}[(V_k - V^\pi)^\top A^\top M w(M_k^\pi, V_k)]$ is not zero in comparison to the i.i.d. sampling update in (4). This is due to the Markovian noise and correlation between $V_k$ and the samples in the replay buffer. As discussed in Section 3.3, the bound on the term $\mathbb{E}[(V_k - V^\pi)^\top A^\top M w(M_k^\pi, V_k)]$ can be decomposed into two parts: the concentration error corresponding to the mini-batch uniformly sampled from the replay buffer, denoted as $E_1^{\text{avg}}$, and the Markovian noise term, denoted as $E_2^{\text{avg}}$. The first part, $E_1^{\text{avg}}$, can be controlled by the size of the mini-batch, decreasing at the order of $O\left(\frac{1}{\sqrt{L}}\right)$, as shown in Lemma 3.3. On the other hand, the Markovian noise term, $E_2^{\text{avg}}$, can be controlled by the size of the replay buffer, decreasing at the order of $O\left(\frac{1}{\sqrt{N}}\right)$. The terms, $E_3^{\text{avg}}$ and $E_4^{\text{avg}}$, arise from the second moment of $\mathbb{E}[||w(M_k, V_k)||_2^2]$. These terms are non-zero in both i.i.d. sampling and Markovian noise cases under the standard TD-learning update (4). However, the errors can be controlled by the size of the mini-batch and replay buffer, as shown in Lemma 3.4. Specifically, $E_3^{\text{avg}}$ and $E_4^{\text{avg}}$ can be decreased at the order of $O\left(\frac{1}{L}\right)$ and $O\left(\frac{1}{N}\right)$, respectively.

Furthermore, the condition $N > \tau^{\text{mix}}$ is standard in the literature in sense that the analysis of Markovian observation model holds after a quantity related to mixing time (Srikant & Ying, 2019; Chen et al., 2022). $\tau^{\text{mix}}$ is only logarithmically proportional to the minimum of the stationary distribution, i.e., $\tau^{\text{mix}} \approx \log\frac{1}{\bar{\mu}_{\min}}$ where $\bar{\mu}_{\min} = \min\{\min_{s,s'\in\mathcal{O}}\mu_{S_\infty,S'_\infty}^\pi(s,s'),\mu_{\min}^\pi\}$. Assuming a uniform transition matrix, i.e., $\mathcal{P}(s,s') = \frac{1}{|\mathcal{S}|}$ for all $s,s' \in \mathcal{S}$, we have $\bar{\mu}_{\min}^\pi = \frac{1}{|\mathcal{S}|^2}$, and we have $\tau^{\text{mix}} \approx 2\log(|\mathcal{S}|)$.

Table 1: Comparitive analysis on results of root mean squared error of averaged iterate convergence using constant step-size.

| Method | Experience replay | Observation model | Step-size | Feature | Initial distribution |
|---|---|---|---|---|---|
| Ours | ✓ | Markovian | $\alpha \in (0,1)$ | Tabular | Arbitrary |
| Bhandari et al. (2018) | ✗ | Markovian | $\frac{1}{\sqrt{T}}$ | Linear | Stationary |
| Lee & Kim (2022) | ✗ | i.i.d. | $\alpha \in (0,1)$ | Tabular | Arbitrary |
| Lakshmi-narayanan & Szepesvari (2018) | ✗ | i.i.d. | Universal | Linear | Arbitrary |

The paper (Bhandari et al., 2018) adopted the assumption that the initial state distribution is already the stationary distribution for simplicity of the proof. Moreover, Lakshminarayanan & Szepesvari (2018) derives a convergence rate of $O(\frac{1}{T})$ for general linear stochastic approximation problems. The universal step-size means that the step-size is dependent on the general linear stochastic approximation problems.

### 3.5 Comparative analysis

Table 1 presents a comprehensive comparison of the finite-time analysis of TD-learning. In the context of on-policy linear function approximation, and under the assumption of starting from the stationary distribution, the work presented in Bhandari et al. (2018) derives the following convergence rate of $\frac{1}{\sqrt{T}}$ for the averaged iterate of TD-learning with a constant step-size, where $T \in \mathbb{N}$ represents the final time of the iterate:

$$\mathbb{E}\left[\left\|\frac{1}{T}\sum_{k=0}^{T}V_k - V^\pi\right\|_2\right] \leq O\left(\sqrt{\frac{\log T}{2(1-\gamma)^3\mu_{\min}^\pi\sqrt{T}}}\right).$$

Notably, our result does not impose any condition on the step-size $\alpha \in (0,1)$, indicating that the use of experience replay memory can ease the strict requirements for selecting a constant step-size. Moreover, we can achieve fast convergence rate while maintaining small constant error by controlling the size of mini-batch and replay buffer instead of controlling the step-size. The result in Lee & Kim (2022) also holds for general step-size condition, but to obtain smaller bias, it requires small step-size yielding a slower convergence rate.

Under the i.i.d. observation model and linear function approximation, Theorem 1 in Lakshminarayanan & Szepesvari (2018) provides $O\left(\frac{1}{\sqrt{T}}\right)$ convergence rate for the root mean squared error using a specific step-size depending on the model parameters, which cannot be known beforehand in TD-learning. The work assumes a more general setup. However, the step-size condition depends on not only the feature space but also $\mu_{\min}^\pi$, the minimum state-visitation distribution even in the tabular case.

### 3.6 Final iterate convergence

In the preceding subsection, we presented a finite-time analysis of the averaged iterate. In this subsection, we extend our analysis to investigate the convergence of the final iterate in TD-learning with a replay buffer, following a similar approach to the one used in the previous section. Instead of using Lyapunov arguments, we utilize the recursive formulas and the fact that $||A||_\infty < 1$ as given in Lemma 3.2. In contrast to the averaged iterate analysis, we assume that the initial distribution corresponds to the stationary distribution of the Markov chain, a common assumption in the literature (Bhandari et al., 2018; Nagaraj et al., 2020; Jain et al., 2021). With the assumption, we are able to derive the convergence rate of the final iterate.

**Theorem 3.6.** *Suppose $S_{-N} \sim \mu_{S_\infty}^\pi$.*

*1) For any $k \geq 0$, we have*

$$\mathbb{E}\left[\|V_k - V^\pi\|_2^2\right] \leq \|V_0 - V^\pi\|_2^2\|\mathcal{S}\|(1 - \alpha(1-\gamma)\mu_{\min}^\pi)^{2k+2}$$

$$+ \underbrace{\frac{64|\mathcal{S}|^2 R_{\max}^2}{(1-\gamma)^3 \mu_{\min}^\pi} \sqrt{\frac{8\log(2|\mathcal{S}|)}{L}}}_{E_1^{\text{final}}: \text{ Concentration error in the first moment}} + \underbrace{\frac{64|\mathcal{S}|^{\frac{7}{2}}|\mathcal{A}|R_{\max}^2}{(1-\gamma)^3 \mu_{\min}^\pi} \sqrt{\frac{\tau^{\text{mix}}}{N}}}_{E_2^{\text{final}}: \text{ Markovian noise in the first moment}}$$

(15)

$$+ \underbrace{\alpha \frac{4|\mathcal{S}|(R_{\max}+1)^2}{(1-\gamma)^3 \mu_{\min}^\pi} \frac{120(\log 2|\mathcal{S}|)^2}{L}}_{E_3^{\text{final}}: \text{ Concentration error in the second moment}} + \underbrace{\alpha \frac{16|\mathcal{S}|^5|\mathcal{A}|^2(R_{\max}+1)^2}{(1-\gamma)^3 \mu_{\min}^\pi} \frac{\tau^{\text{mix}}}{N}}_{E_4^{\text{final}}: \text{ Markovian noise in the second moment}} .$$

*2) For any $k \geq 0$, we have*

$$\mathbb{E}\left[\|V_k - V^\pi\|_2\right] \leq \sqrt{|\mathcal{S}|} \|V_0 - V^\pi\|_2 (1 - \alpha(1-\gamma)\mu_{\min}^\pi)^{k+1}$$
$$+ \frac{8|\mathcal{S}|R_{\max}}{(1-\gamma)^{\frac{3}{2}}(\mu_{\min}^\pi)^{\frac{1}{2}}} \sqrt{\sqrt{\frac{8\log(2|\mathcal{S}|)}{L}} + 2|\mathcal{S}|^{\frac{3}{2}}|\mathcal{A}|\sqrt{\frac{\tau^{\text{mix}}}{N}}}$$
$$+ \sqrt{\alpha} \frac{2|\mathcal{S}|^{\frac{1}{2}}(R_{\max}+1)}{(1-\gamma)^{\frac{3}{2}}(\mu_{\min}^\pi)^{\frac{1}{2}}} \sqrt{\frac{120(\log(2|\mathcal{S}|)^2}{L} + 4|\mathcal{S}|^4|\mathcal{A}|^2 \frac{\tau^{\text{mix}}}{N}}.$$

(16)

The proof is given in Appendix Section A.10. In a manner similar to the case of the averaged iterate, the term (15) arises due to the non-zero value of $\mathbb{E}[(V_k - V^\pi)^\top A^\top M w(M_k^\pi, V_k)]$ as compared to the i.i.d. sampling update with (4). This non-zero value is caused by Markovian noise and correlation between $V_k$ and the samples in the replay buffer. As explained in Section 3.3, the bound on the term $\mathbb{E}[(V_k - V^\pi)^\top A^\top M w(M_k^\pi, V_k)]$ can be decomposed into two parts: $E_1^{\text{final}}$, which is the concentration error of uniform random sampling, and $E_2^{\text{final}}$, which is the Markovian noise term. As can be seen from Lemma 3.3, $E_1^{\text{final}}$ can be controlled by the size of mini-batch, where it can be decreased at the order of $O\left(\frac{1}{\sqrt{L}}\right)$. Moreover, the Markovian noise $E_2^{\text{final}}$ can be controlled by the replay buffer size, decreasing at the order of $O\left(\frac{1}{\sqrt{N}}\right)$. The error bounds provided in above are new in the sense that the it shows dependency on the size of the replay buffer which can be controlled by adjusting the buffer size, $L$ and $N$.

Note that the assumption on the initial distribution is for simplicity of the proof, and it was also used in Bhandari et al. (2018). The assumption can be relaxed to more practical case that the initial distribution is arbitrary with some more efforts. To this end, we can bound the error caused by the initial distribution using the condition in (2). As from Bhandari et al. (2018), we can bound the error by $O(\alpha)$, which will ultimately result in a $O(\alpha)$ in the final error bound. However, such error is dominated by the $O(\alpha^{\frac{1}{2}})$ term in the error bound. Therefore, it does not significantly impact the convergence rate or the sample complexity.

### 3.7 Comparative analysis

The overall comparative analysis is given in Table 2. Under the Markovian assumption and on-policy linear function approximation, Bhandari et al. (2018) provided a final iterate convergence under the constant step-size $\alpha$ smaller than $\frac{1}{\mu_{\min}^\pi(1-\gamma)}$, which is

$$\mathbb{E}\left[\|V_T - V^\pi\|_2\right] \leq \left(e^{-\alpha(1-\gamma)\mu_{\min}^\pi T}\right) \|V^\pi - V_0\|_2 + \underbrace{\sqrt{\alpha \left(\frac{(9 + 12t^{\text{mix}}(\alpha))}{2(1-\gamma)^3 \mu_{\min}^\pi}\right)}}_{\text{Error from constant step-size}}.$$

(17)

Although the constant step-size is a widely used method, it incurs constant error terms. However, the use of experience replay can reduce these errors. Specifically, as stated in the second statement of Theorem 3.6, the term (16) decreases at the order of $O\left(\frac{1}{L^{\frac{1}{4}}} + \frac{1}{N^{\frac{1}{4}}}\right)$. Furthermore, as in the averaged iterate case, since we do not impose any condition on the step-size, we can achieve fast convergence rate while maintaining small error by controlling the size of mini-batch and replay buffer instead of the step-size.

The work in Lee & Kim (2022) provided a final iterate convergence of tabular TD-learning with a constant step-size and i.i.d. observation models, where the step-size induces constant errors proportional to $O(\sqrt{\alpha})$.

Table 2: Comparitive analysis on results of root mean squared error of final iterate convergence using constant step-size

| Method | Experience replay | Observation model | Step-size | Feature | Constant error term |
|---|---|---|---|---|---|
| Ours | ✓ | Markovian | $\alpha \in (0,1)$ | Tabular | $O\left(\frac{1}{L^{\frac{1}{4}}} + \frac{\tau^{\mathrm{mix}}}{N^{\frac{1}{4}}}\right)$ |
| Bhandari et al. (2018) | ✗ | Markovian | Model dependent | Linear | $O\left(\sqrt{\alpha \log(\alpha)}\right)$ |
| Lee & Kim (2022) | ✗ | i.i.d. | $\alpha \in (0,1)$ | Tabular | $O(\sqrt{\alpha})$ |
| Srikant & Ying (2019) | ✗ | Markovian | Model dependent | Linear | $O(\sqrt{\alpha})$ |
| Dalal et al. (2018) | ✗ | i.i.d. | $\frac{1}{(k+1)^\sigma}$, $\sigma \in (0,1)$ | Linear | $O(\frac{1}{T^\sigma})$, $\sigma \in (0,1)$ |

Model dependent step size implies that the step size depends on model parameters, e.g., maximum eignevalue of matrix $A$, discount factor $\gamma$, mixing time.

The approach in Srikant & Ying (2019) provided a mean squared bound of TD-learning under linear function scheme and Markovian observation models, which is given by $\mathbb{E}\left[\|V_T - V^\pi\|_2^2\right] \leq O\left((1 - \alpha c_1)^{T-\tau} + c_2\alpha\tau\right)$ where $c_1, c_2$ are model dependent parameters, and $\tau$ is the mixing time such that $\forall k \geq \tau$, $\left\|\gamma D^\pi P^\pi - D^\pi - (\mathbb{E}\left[\gamma e_{s_k}^\top e_{s_{k+1}} - e_{s_k}^\top e_{s_k}\right])\right\|_2 \leq \delta$ and $\left\|D^\pi R^\pi - \mathbb{E}\left[e_{s_k}^\top R^\pi\right]\right\|_2 \leq \delta$. However, the choice of the step-size depends on the mixing time and the model parameters.

Under the i.i.d. assumption and using linear function approximation, Dalal et al. (2018) derived $O\left(\frac{1}{T^\sigma}\right)$, $\sigma \in (0,1)$ bounds on the mean squared error bound, which is worse than $O\left(\frac{1}{T}\right)$ convergence rate. Since we used constant step-size, the result is not directly comparable.

Furthermore, our approach addresses a scenario that standard TD-learning analysis cannot : Suppose starting with a pre-existing dataset before transitioning to online learning, a topic of recent interest (Song et al., 2022). Then, the convergence rate, $\mathcal{O}(\exp(-\alpha k))$, becomes faster as we choose large $\alpha$, which is possible thanks to $\alpha \in (0,1)$. This will only require $\tilde{\mathcal{O}}\left(\frac{1}{(1-\gamma)\mu_{\min}^\pi} \frac{\ln\left(\frac{1}{\epsilon}\right)}{\alpha}\right)$ number of samples for $\mathbb{E}\left[\|V_k - V^\pi\|\right] \leq \epsilon$ to hold for $\epsilon > 0$, which is the so-called sample complexity. In other words, if we exclude $N$ in the sample complexity, permitting large $\alpha$ yields better sample complexity. However, as discussed in Appendix A.11, if we take into account $N$ in the sample complexity, the overall sample complexity $\tilde{\mathcal{O}}\left(\frac{|\mathcal{S}|^7|\mathcal{A}|^2\tau^{\mathrm{mix}}}{(1-\gamma)^6(\mu_{\min}^\pi)^4}\frac{1}{\alpha}\frac{1}{\epsilon^4}\right)$ can be sub-optimal in terms of $\frac{1}{1-\gamma}$ an $\mu_{\min}^\pi$, compared to the result of $\tilde{\mathcal{O}}\left(\frac{1}{\epsilon^2}\frac{t^{\mathrm{mix}}}{(1-\gamma)^4(\mu_{\min}^\pi)^2}\right)$ by Bhandari et al. (2018).

Moreover, introducing $L$ and $N$ can relax the condition on $\alpha$ for the convergence in expectation to hold. The related works require condition on $\alpha$ for the convergence statement to hold. However, we do not require such conditions, while the error can be controlled by $L$ and $N$. Moreover, the convergence statement holds for arbitrary $\alpha \in (0,1)$, $L$, and $N$ where $N$ only dependes on the mixing time, $\tau^{\mathrm{mix}}$.

Lastly, we would like to emphasize that our main contributions are as providing the first analysis of reinforcement learning in a setting that closely resembles the deep Q-learning approach described in Mnih et al. (2015) within the TD-learning framework, and providing a rigorous analysis on the relation between the error bound and the mini-batch size ($L$), and the replay buffer size ($N$) within the TD-learning framework. We hope these contributions will provide valuable insights to the community. In particular, the error bounds given in our paper are new in the sense that the it depends on the size of the replay buffer which can be controlled by adjusting the buffer size. For instance, the constant error ball caused by the constant step-size can become arbitrarily small by increasing the buffer size.

## 4 Conclusion

In this work, we have undertaken an analysis of the behavior of TD-learning utilizing experience replay memory under a Markovian observation model, which has thus far been unexplored despite the prevalence of experience replay memory in RL algorithms. By leveraging a simple matrix concentration inequality and the geometric mixing property of irreducible and aperiodic Markov chains, we have demonstrated that the expected root mean squared error of the averaged iterate of TD-learning can be reduced at the order of $O\left(\frac{1}{L^{\frac{1}{4}}} + \frac{1}{N^{\frac{1}{4}}}\right)$ under the constant step-size. Similarly, for the final iterate case, we have established that the root mean squared error can be effectively reduced at the order of $O\left(\frac{1}{L^{\frac{1}{4}}} + \frac{1}{N^{\frac{1}{4}}}\right)$. Potential avenues for future research include extending the proposed analysis frameworks to off-policy and linear function approximation settings, as well as investigating the applicability of these results to Q-learning.

## 5 Acknowledgement

The work was supported in part by the BK21 FOUR from the Ministry of Education (Republic of Korea), and in part by the Institute of Information Communications Technology Planning Evaluation (IITP) funded by the Korea government under Grant 2022-0-00469. The work was supported by KT Corporation as part of the KT-KAIST Open R&D project.

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

# A  Appendix

## A.1  Notations

Throughout the paper, the following notations will be adopted: $\mathbb{R}$: set of real numbers; $\mathbb{R}^n$: set of all $n$-dimensional vectors; $\mathbb{R}^{n \times m}$: set of all $n \times m$ real matrices; $\mathbb{Z}_+$: set of all non-negative integers; $\mathbb{R}_+$ : set of non-negative real numbers; $\mathbb{R}_+$ : set of non-negative real numbers; for matrix $A \in \mathbb{R}^{n \times m}$, $[A]_{ij}$, $1 \le i \le n, 1 \le j \le m$ denotes $i$-th row and $j$-th column element of $A$; $e_s \in \mathbb{R}^n$ for $1 \le s \le n$ : $s$-th basis vector of $\mathbb{R}^{|\mathcal{S}|}$ space, i.e., only the $s$-th element of $e_s$ is one and other elements are zero; $||A||_\infty$ for $A \in \mathbb{R}^{n \times m}$ denotes the infinity norm $||A||_\infty := \max_{1 \le i \le n} \sum_{j=1}^m |[A]_{ij}|$ ; $|S|$: cardinality of a finite set $S$; $O(\cdot)$ denotes the big O notation.

## A.2  Technical Lemmas

**Lemma A.1.** *[Concentration bound for i.i.d. matrix random variables (Tropp et al. (2015),Corollary 6.2.1)] Let $X \in \mathbb{R}^{d_1 \times d_2}$, where $d_1$ and $d_2$ are some positive integers. Moreover, assume that the sequence of random*

matrices $\{X_k\}_{k=1}^n$ are i.i.d. samples from a distribution such that $\mathbb{E}[X_k] = X$ and $||X_k||_2 \leq X_{\max}$ for all $1 \leq k \leq n$. Let $\sigma = ||\mathbb{E}[X_k X_k^\top]||_2$. Then, we get

$$\mathbb{P}\left[\left\|\frac{1}{n}\sum_{k=1}^n X_k - X\right\|_2 \geq t\right] \leq (d_1 + d_2)\exp\left(-nt^2/(\sigma + 2X_{\max}t/3)\right),$$

and

$$\mathbb{E}\left[\left\|\frac{1}{n}\sum_{k=1}^n X_k - X\right\|_2\right] \leq \sqrt{\frac{2\sigma \log(d_1 + d_2)}{n}} + \frac{2L\log(d_1 + d_2)}{3n}.$$

With the above result, a bound on the second moment $\mathbb{E}\left[\left\|\frac{1}{n}\sum_{k=1}^n X_k - X\right\|_2^2\right]$ can be obtained as follows.

**Corollary A.2.** *Let $X \in \mathbb{R}^{d_1 \times d_2}$. Assume that the sequence of random matrices $\{X_k\}_{k=1}^n$ are i.i.d. samples from a distribution such that $\mathbb{E}[X_k] = X$ and $||X_k||_2 \leq X_{\max}$ for all $1 \leq k \leq n$. Letting $||\mathbb{E}[X_k X_k^\top]||_2 \leq \sigma$, the corresponding second moment can be bounded as follows:*

$$\mathbb{E}\left[\left\|\frac{1}{n}\sum_{k=1}^n X_k - X\right\|_2^2\right]$$
$$\leq \frac{2\sigma}{n}\log(d_1 + d_2) + \frac{2\sigma}{n} + \frac{16X_{\max}^2}{9n^2}(\log(d_1 + d_2))^2 + \frac{32X_{\max}^2}{9n^2}\log(d_1 + d_2) + \frac{8X_{\max}}{3n}.$$

*Proof.* The proof is completed by the inequalities

$$\mathbb{E}\left[\left\|\frac{1}{n}\sum_{k=1}^n X_k - X\right\|_2^2\right]$$
$$= \int_0^\infty \mathbb{P}\left[\left\|\frac{1}{n}\sum_{k=1}^n X_k - X\right\|_2 \geq \sqrt{t}\right] dt$$
$$\leq \int_0^\infty \min\left\{(d_1 + d_2)\exp(-nt/(\sigma + 2X_{\max}\sqrt{t}/3)), 1\right\} dt$$
$$\leq \int_0^\infty \min\left\{1, (d_1 + d_2)\exp(-nt/(2\sigma))\right\} dt + \int_0^\infty \min\left\{1, (d_1 + d_2)\exp(-3n\sqrt{t}/(4X_{\max}))\right\} dt$$
$$\leq \frac{2\sigma}{n}\log(d_1 + d_2) + \frac{2\sigma}{n} + \frac{16X_{\max}^2}{9n^2}(\log(d_1 + d_2))^2 + \int_{\frac{16X_{\max}^2}{9n^2}(\log(d_1+d_2))^2}^\infty (d_1 + d_2)\exp(-3n\sqrt{t}/(4X_{\max}))dt$$
$$\leq \frac{2\sigma}{n}\log(d_1 + d_2) + \frac{2\sigma}{n} + \frac{16X_{\max}^2}{9n^2}(\log(d_1 + d_2))^2 + \int_{\frac{4X_{\max}}{3n}\log(d_1+d_2)}^\infty 2(d_1 + d_2)u\exp(-3nu/(4X_{\max}))du$$
$$\leq \frac{2\sigma}{n}\log(d_1 + d_2) + \frac{2\sigma}{n} + \frac{16X_{\max}^2}{9n^2}(\log(d_1 + d_2))^2 + \frac{32X_{\max}^2}{9n^2}\log(d_1 + d_2) + \frac{8X_{\max}}{3n},$$

where the first equality follows from the inequality, $\mathbb{E}[Y] = \int_0^\infty \mathbb{P}[Y \geq t]\,dt$ for non-negative random variable $Y$, which can be found in Exercise 2.14 in Casella & Berger (2021).

Moreover, the first inequality follows from applying Lemma A.1, and the last inequality follows from applying change of variables $\sqrt{t} = x$ and integration by parts. $\qquad\square$

**Lemma A.3** (Variance of empirical distribution, Paulin (2015), Proposition 3.21)**.** *Let $\{S_k\}_{k=0}^n$ be a Markov chain following transition kernel $\mathcal{P}^\pi$ with the state space $\mathcal{S}$ and starting from its stationary distribution $\mu_{S_\infty}^\pi$, i.e., $S_0 \sim \mu_{S_\infty}^\pi$. The empirical distribution is defined as $\mu^{\mathrm{em}}(s) := \sum_{k=1}^n \mathbf{1}\{S_k = s\}/n$, $s \in \mathcal{S}$. Then,*

*1)* $\sum_{s \in \mathcal{S}} \mathbb{E}\left[(\mu_{S_\infty}^\pi(s) - \mu^{\mathrm{em}}(s))^2\right] \leq |\mathcal{S}|\frac{t^{\mathrm{mix}}}{n},$

*2)* $\mathbb{E}\left[d_{\mathrm{TV}}(\mu_{S_\infty}^\pi, \mu^{\mathrm{em}}) \leq\right] \leq |\mathcal{S}|\sqrt{\frac{t^{\mathrm{mix}}}{n}}$.

**Lemma A.4** (Bound on total variation distance, Chapter 4.5 and Exercise 4.2 in Levin & Peres (2017))**.** *Let $\{S_k\}_{k=0}^n$ be a Markov chain following transition kernel $\mathcal{P}^\pi$ with the state space $\mathcal{S}$. Denote its stationary distribution $\mu_{S_\infty}^\pi$ and mixing time as $t^{\mathrm{mix}}$. Then, the following holds:*

*1) For a non-negative integer $l$, we have $\sup_{s \in \mathcal{S}} d_{\mathrm{TV}}(\mathbb{P}[S_{lt^{\mathrm{mix}}} \mid S_0 = s], \mu_{S_\infty}^\pi) \leq 2^{-l}$.*

*2) $\sup_{s \in \mathcal{S}} d_{\mathrm{TV}}(\mathbb{P}[S_k \mid S_0 = s], \mu_{S_\infty}^\pi)$ is non-increasing in terms of $k$.*

**Lemma A.5** (Proposition 3.4 in Paulin (2015))**.** *For irreducible and aperiodic Markov chain, we have, for $\epsilon > 0$*

$$t^{\mathrm{mix}}(\epsilon) \leq t^{\mathrm{mix}}\left(\frac{1}{4}\right)\left(1 + 2\log\left(\frac{1}{\epsilon}\right) + \log\left(\frac{1}{d_{\min}}\right)\right).$$

### A.3 Proof of Lemma 2.3

Before the main proof, the formal definition of irreducible and aperiodic Markov chain is reviewed first.

**Definition A.6** (Irreducible and aperiodic Markov chain (Levin & Peres, 2017))**.** *A Markov chain is called irreducible if for any two states $x, y \in \mathcal{X}$, $e_x^\top P^k e_y > 0$ for some $k \geq 0$. Let $\mathcal{T}(x) := \{k \geq 1 \mid e_x^\top P^k e_x > 0\}$ be the set of time instances such that the probability of returning back to the initial state $x$ is positive. The period of state $x$ is the greatest common divisor of $\mathcal{T}(x)$. A Markov chain is said to be aperiodic if all states have period one.*

The proof of Lemma 2.3 is given below.

*Proof.* First of all, we prove that the extended Markov chain is irreducible. By hypothesis, the Markov chain, $(S_k)_{k=0}^\infty$, with transition kernel $\mathcal{P}^\pi$ is irreducible. Then, one can prove that starting from any $(S_k, S_{k+1}) = (x, y) \in \mathcal{O}$, there exists a positive probability that any $(S_{k+T}, S_{k+1+T}) = (w, z) \in \mathcal{O}$ can be reached for some $T \geq 0$. This is because by hypothesis, $\mathbb{P}[S_{k+T} = w \mid S_0 = x] > 0$ for some $T \geq 0$, and since $(w, z) \in \mathcal{O}$ is a realizable pair, $(S_{k+T}, S_{k+1+T}) = (w, z) \in \mathcal{O}$ is visited with a positive probability. This proves the irreducibility.

Next, we prove the aperiodicity. By hypothesis, the Markov chain, $(S_k)_{k=0}^\infty$, is aperiodic. To prove that the induced Markov chain $(S_k, S_{k+1})_{k=0}^\infty$ is aperiodic by contradiction, let us suppose that $(S_k, S_{k+1})_{k=0}^\infty$ is periodic with some period $T > 1$. Then, there exists a a state $s \in \mathcal{S}$ such that its period is larger than one, which contradicts the aperiodicity assumption of $(S_k)_{k=0}^\infty$. This completes the proof.

$\square$

### A.4 Proof of Lemma 3.1

*Proof.* The proof proceeds by induction. Suppose $||V_k||_\infty \leq \frac{R_{\max}}{1-\gamma}$ holds for $k \geq 0$. Rewriting (6), we have

$$V_{k+1} = \left(I + \alpha\frac{1}{|M_k^\pi|}\sum_{(s,r,s')\in M_k^\pi}(\gamma e_s e_{s'}^\top - e_s e_s^\top)\right)V_k + \alpha\frac{1}{|M_k^\pi|}\sum_{(s,r,s')\in M_k^\pi}e_s r.$$

We can prove that for $s \in \mathcal{S}$, if $(s, r, s') \notin M_k^\pi$, then $e_s^\top V_{k+1}$ is identical to $e_s^\top V_k$. If $(s, r, s') \in M_k^\pi$ for some $s \in \mathcal{S}$, then the update of $e_s^\top V_{k+1}$ can be expressed as follows:

$$e_s^\top V_{k+1} = e_s^\top\left\{\left(I + \alpha\frac{1}{|M_k^\pi|}\sum_{(s,r,s')\in M_k^\pi}(\gamma e_s e_{s'}^\top - e_s e_s^\top)\right)V_k + \alpha\frac{1}{|M_k^\pi|}\sum_{(s,r,s')\in M_k^\pi}e_s r\right\}.$$

Consider a subset of the mini-batch $M_k^\pi$ such that the first element of the tuple is $s$, i.e., $M_k^\pi(s) := \{(\tilde{s}, \tilde{r}, \tilde{s}') \in M_k^\pi : \tilde{s} = s\} \subseteq M_k^\pi$. Then, we can bound $e_s^\top V_{k+1}$ as follows:

$$
\begin{aligned}
|e_s^\top V_{k+1}| &\le \left| e_s^\top \left( I + \alpha \frac{1}{|M_k^\pi|} \sum_{(s,r,s') \in M_k^\pi(s)} (\gamma e_s e_{s'}^\top - e_s e_s^\top) \right) V_k \right| + \alpha \frac{|M_k^\pi(s)|}{|M_k^\pi|} R_{\max} \\
&\le \left\| \left( I + \alpha \frac{1}{|M_k^\pi|} \sum_{(s,r,s') \in M_k^\pi(s)} (\gamma e_s e_{s'}^\top - e_s e_s^\top) \right)^\top e_s \right\|_1 \|V_k\|_\infty + \alpha \frac{|M_k^\pi(s)|}{|M_k^\pi|} R_{\max} \\
&\le \left( 1 + \alpha \frac{|M_k^\pi(s)|}{|M_k^\pi|} (\gamma - 1) \right) \|V_k\|_\infty + \alpha \frac{|M_k^\pi(s)|}{|M_k^\pi|} R_{\max} \\
&\le \frac{R_{\max}}{1 - \gamma},
\end{aligned}
$$

where the first inequality follows from the application of the triangle inequality and the assumption that the reward is bounded, as given in Assumption 2.4. The second inequality is due to the Cauchy-Schwartz inequality, and the last inequality comes from the induction hypothesis. This completes the proof. $\qquad\square$

### A.5 Proof of Lemma 3.2

*Proof.* First of all, to prove the first statement, $||A||_\infty$ is bounded as

$$
||A||_\infty = \max_{1 \le i \le |\mathcal{S}|} \left( 1 - \alpha [D^\pi]_{ii} + \alpha \gamma [D^\pi] \sum_{j=1}^{|\mathcal{S}|} [P^\pi]_{ij} \right) = \min_{s \in \mathcal{S}} (1 - \alpha(1 - \gamma) \mu^\pi(s)),
$$

where the last equality is obtained by expanding $A$ according to the definition (8) from the fact that $P^\pi$ is a stochastic matrix, i.e., the row sum of the matrix equals one.

Next, for the second statement, one can readily prove that $M := \sum_{k=0}^{\infty} (A^k)^\top (A)^k$ is a solution of (10) by simply plugging it into $M$ in (10). It remains to prove that $M$ is bounded. In particular, $||M||_2$ is bounded as

$$
\begin{aligned}
||M||_2 &\le ||I||_2^2 + ||A||_2^2 + ||A||_2^4 + \cdots \\
&\le 1 + |\mathcal{S}| ||A||_\infty^2 + |\mathcal{S}| ||A||_\infty^4 + \cdots \\
&\le 1 + |\mathcal{S}| (1 - \alpha(1 - \gamma)\mu_{\min}^\pi)(1 + (1 - \alpha(1 - \gamma)\mu_{\min}^\pi)^2 + (1 - \alpha(1 - \gamma)\mu_{\min}^\pi)^4 + \cdots \\
&\le 1 + \frac{|\mathcal{S}|}{\alpha(1 - \gamma)\mu_{\min}^\pi},
\end{aligned}
$$

where the first inequality follows from applying triangle inequality and the fact that $||A^\top||_2 = ||A||_2$. The second inequality is due to the relation $||A||_2 \le \sqrt{|\mathcal{S}|} ||A||_\infty$, and the third inequality comes from Lemma 3.2. $\qquad\square$

### A.6 Auxiliary lemmas to prove Lemma 3.3

Let $\mu_{\tilde{S}}^{\tilde{B}_k^\pi}$ and $\mu_{\tilde{S},\tilde{S}'}^{\tilde{B}_k^\pi}$ be the empirical distributions of the state $s$ and the consecutive state pair $(s, s')$ in the replay buffer, $\tilde{B}_k^\pi$, respectively, which are defined in the same manner as $\mu_S^{B_k^\pi}$ and $\mu_{S,S'}^{B_k^\pi}$, respectively.

**Lemma A.7.** *For $k \ge 0$, we have the following bounds:*

*1)* $\mathbb{E}\left[ \left\| D^\pi - D^{\tilde{B}_k^\pi} \right\|_2 \right] \le \sqrt{|\mathcal{S}|} \sqrt{\frac{t_1^{\mathrm{mix}}}{|\tilde{B}_k^\pi|}},$

*2)* $\mathbb{E}\left[ \left\| D^\pi P^\pi - D^{\tilde{B}_k^\pi} P^{\tilde{B}_k^\pi} \right\|_2 \right] \le |\mathcal{S}| \sqrt{\frac{t_2^{\mathrm{mix}}}{|\tilde{B}_k^\pi|}},$

*3)* $\mathbb{E}\left[\left\|D^{\pi}R^{\pi} - D^{\tilde{B}_k^{\pi}}R^{\tilde{B}_k^{\pi}}\right\|_2\right] \leq |\mathcal{S}|^{\frac{5}{2}}|\mathcal{A}|R_{\max}\sqrt{\frac{t_2^{\mathrm{mix}}}{|\tilde{B}_k^{\pi}|}}.$

*Proof.* First of all, the expected value of $D^{\pi} - D^{\tilde{B}_k^{\pi}}$ can be bounded as follows:

$$\mathbb{E}\left[\left\|D^{\pi} - D^{\tilde{B}_k}\right\|_2\right] \leq \mathbb{E}\left[\sqrt{\sum_{s\in\mathcal{S}}\left(\mu_{S_{\infty}}^{\pi}(s) - \mu_{\tilde{S}}^{\tilde{B}_k^{\pi}}(s)\right)^2}\right] \leq \sqrt{|\mathcal{S}|}\sqrt{\frac{t_1^{\mathrm{mix}}}{|\tilde{B}_k^{\pi}|}},$$

which proves the first statement. In the above inequalities, the first inequality follows from the fact that $||A||_2 \leq ||A||_F$, where $A \in \mathbb{R}^{|\mathcal{S}|\times|\mathcal{S}|}$ and $||\cdot||_F$ stands for Frobenius norm. The last inequality follows from Jensen's inequality and Lemma A.3 in Appendix.

The second statement can be proved using the following similar arguments:

$$\mathbb{E}\left[\left\|D^{\pi}P^{\pi} - D^{\tilde{B}_k^{\pi}}P^{\tilde{B}_k^{\pi}}\right\|_2\right] \leq \mathbb{E}\left[\sqrt{\sum_{(s,s')\in\mathcal{S}\times\mathcal{S}}\left(\mu_{S_{\infty},S_{\infty}'}^{\pi}(s,s') - \mu_{\tilde{S},\tilde{S}'}^{\tilde{B}_k^{\pi}}(s,s')\right)^2}\right] \leq |\mathcal{S}|\sqrt{\frac{t_2^{\mathrm{mix}}}{|\tilde{B}_k^{\pi}|}},$$

where the first inequality follows from the fact that $||A||_2 \leq ||A||_F$, where $A \in \mathbb{R}^{|\mathcal{S}|\times|\mathcal{S}|}$ and $||\cdot||_F$, the second inequality is due to Lemma A.3 in Appendix, and follows from applying Jensen's inequality.

For the third statement, we can bound $\mathbb{E}\left[\left\|D^{\pi}R^{\pi} - D^{\tilde{B}_k^{\pi}}R^{\tilde{B}_k^{\pi}}\right\|_2\right]$ in a similar sense as follows:

$$\mathbb{E}\left[\left\|D^{\pi}R^{\pi} - D^{\tilde{B}_k^{\pi}}R^{\tilde{B}_k^{\pi}}\right\|_2\right] \leq \mathbb{E}\left[\sqrt{\sum_{s\in\mathcal{S}}\left(\mathbb{E}\left[r(S,A,S')\mid S=s,\pi\right]\mu_{S_{\infty}}^{\pi}(s) - \mu_{\tilde{S}}^{\tilde{B}_k^{\pi}}(s)[R^{\tilde{B}_k^{\pi}}]_s\right)^2}\right]$$

$$\leq |\mathcal{S}||\mathcal{A}|R_{\max}\mathbb{E}\left[\sqrt{\sum_{s\in\mathcal{S}}\left(\sum_{s'\in\mathcal{S}}\mu_{S_{\infty},S_{\infty}'}^{\pi}(s,s') - \mu_{\tilde{S},\tilde{S}'}^{\tilde{B}_k^{\pi}}(s,s')\right)^2}\right]$$

$$\leq |\mathcal{S}|^{\frac{5}{2}}|\mathcal{A}|R_{\max}\sqrt{\frac{t_2^{\mathrm{mix}}}{|\tilde{B}_k^{\pi}|}},$$

where the second inequality applies the inequality $(\sum_{k=0}^n a_k b_k) \leq (\sum_{k=0}^n a_k)(\sum_{k=0}^n b_k)$ for $a_k, b_k \in \mathbb{R}$, $k \geq 0$, and the relation between $\mu_{S_{\infty}}^{\pi}$ and $\mu_{S_{\infty},S_{\infty}'}^{\pi}$ in (3), and

$$\sum_{a\in\mathcal{A}}\frac{\sum_{l=k-N+1}^{N}\mathbf{1}\{s_l, a_l, s_{l+1} = (s,a,s')\}}{\sum_{l=k-N+l}^{N}\mathbf{1}\{s_l = i\}}\mu_{\tilde{S}}^{\tilde{B}_k^{\pi}}(s) = \mu_{\tilde{S},\tilde{S}'}^{\tilde{B}_k^{\pi}}(s,s').$$

Moreover, the last inequality follows the same logic as that used for bounding $\mathbb{E}\left[\left\|D^{\pi}P^{\pi} - D^{\tilde{B}_k^{\pi}}P^{\tilde{B}_k^{\pi}}\right\|_2\right]$. This completes the proof. $\square$

**Lemma A.8.** *For $k \geq 0$, we have*

*1)* $\mathbb{E}\left[\left\|D^{\pi} - D^{B_k^{\pi}}\right\|_2\right] \leq \sqrt{|\mathcal{S}|}\sqrt{\frac{t_1^{\mathrm{mix}}}{|B_k^{\pi}|}} + 4d_{\mathrm{TV}}(\mu_{S_{\infty}}^{\pi}, \mu_{S_{k-N}}^{\pi}),$

*2)* $\mathbb{E}\left[\left\|D^{\pi}P^{\pi} - D^{B_k^{\pi}}P^{B_k^{\pi}}\right\|_2\right] \leq |\mathcal{S}|\sqrt{\frac{t_2^{\mathrm{mix}}}{|B_k^{\pi}|}} + 4\sqrt{|\mathcal{S}|}d_{\mathrm{TV}}(\mu_{S_{\infty}}^{\pi}, \mu_{S_{k-N}}^{\pi}),$

*3)* $\mathbb{E}\left[\left\|D^{\pi}R^{\pi} - D^{B_k^{\pi}}R^{B_k^{\pi}}\right\|_2\right] \leq |\mathcal{S}|^{\frac{5}{2}}|\mathcal{A}|R_{\max}\sqrt{\frac{t_2^{\mathrm{mix}}}{|B_k^{\pi}|}} + 2\sqrt{|\mathcal{S}|}R_{\max}d_{\mathrm{TV}}(\mu_{S_{\infty}}^{\pi}, \mu_{S_{k-N}}^{\pi}).$

*Proof.* Let $\{\tilde{S}_k\}_{k \geq -N}$ be the stationary Markov chain defined in Section 3.3. The expected value of $\|D^\pi - D^{B_k^\pi}\|_2$ can be bounded using irreducible and aperiodic property of the Markov chain as follows:

$$
\begin{aligned}
\mathbb{E}\left[\left\|D^\pi - D^{B_k^\pi}\right\|_2\right] &= \sum_{s \in \mathcal{S}} \mathbb{E}\left[\left\|D^\pi - D^{B_k^\pi}\right\|_2 \Big| S_{k-N} = s\right] \mathbb{P}[S_{k-N} = s] \\
&= \sum_{s \in \mathcal{S}} \mathbb{E}\left[\left\|D^\pi - D^{\tilde{B}_k^\pi}\right\|_2 \Big| \tilde{S}_{k-N} = s\right] \mathbb{P}[\tilde{S}_{k-N} = s] \\
&\quad + \sum_{s \in \mathcal{S}} \mathbb{E}\left[\left\|D^\pi - D^{\tilde{B}_k^\pi}\right\|_2 \Big| \tilde{S}_{k-N} = s\right] \left(\mathbb{P}[S_{k-N} = s] - \mathbb{P}[\tilde{S}_{k-N} = s]\right) \\
&\leq \mathbb{E}\left[\left\|D^\pi - D^{\tilde{B}_k^\pi}\right\|_2\right] + 2\sum_{s \in \mathcal{S}} |\mathbb{P}[S_{k-N} = s] - \mathbb{P}[\tilde{S}_{k-N} = s]| \\
&\leq \sqrt{|\mathcal{S}|}\sqrt{\frac{t_1^{\mathrm{mix}}}{|B_k^\pi|}} + 4d_{\mathrm{TV}}(\mu_{S_\infty}^\pi, \mu_{S_{k-N}}^\pi).
\end{aligned}
$$

In the above inequalities, the first equality follows from the law of total expectation, and the second equality is due to the fact that $\mathbb{P}[S_{k-N+1}, S_{k-N+2}, \ldots, S_k \mid S_{k-N}] = \mathbb{P}[\tilde{S}_{k-N+1}, \tilde{S}_{k-N+2}, \ldots, \tilde{S}_k \mid \tilde{S}_{k-N}]$ because the transition probabilities of both Markov chains are identical. Next, the first inequality follows from the fact that $\|D^\pi\|_2$ and $\|D^{\tilde{B}_k^\pi}\|_2$ are both smaller than one, and the last equality follows from the definition of the total variation distance in (2.1), and applying Lemma A.7.

In the second statement, the expected value of $\|D^\pi P^\pi - D^{B_k^\pi} P^{B_k^\pi}\|_2$ can be bounded using the same argument as in the first statement. In particular, we have

$$
\begin{aligned}
&\mathbb{E}\left[\left\|D^\pi P^\pi - D^{B_k^\pi} P^{B_k^\pi}\right\|_2\right] \\
&= \sum_{s \in \mathcal{S}} \mathbb{E}\left[\left\|D^\pi P^\pi - D^{\tilde{B}_k^\pi} P^{\tilde{B}_k^\pi}\right\|_2 \Big| \tilde{S}_{k-N} = s\right] \mathbb{P}[\tilde{S}_{k-N} = s] \\
&\quad + \sum_{s \in \mathcal{S}} \mathbb{E}\left[\left\|D^\pi P^\pi - D^{\tilde{B}_k^\pi} P^{\tilde{B}_k^\pi}\right\|_2 \Big| \tilde{S}_k = s\right] \left(\mathbb{P}[S_{k-N} = s] - \mathbb{P}[\tilde{S}_{k-N} = s]\right) \\
&\leq \sqrt{|\mathcal{S}|}\sqrt{\frac{t_2^{\mathrm{mix}}}{N}} + 4\sqrt{|\mathcal{S}|}d_{\mathrm{TV}}(\mu_{S_\infty}^\pi, \mu_{S_{k-N}}^\pi).
\end{aligned}
$$

The same logic holds for $\mathbb{E}\left[\|D^\pi R^\pi - D^{B_k^\pi} R^{B_k^\pi}\|_2\right]$. In particular, it follows from

$$
\begin{aligned}
&\mathbb{E}\left[\left\|D^\pi R^\pi - D^{B_k^\pi} R^{B_k^\pi}\right\|_2\right] \\
&= \sum_{s \in \mathcal{S}} \mathbb{E}\left[\left\|D^\pi R^\pi - D^{\tilde{B}_k^\pi} R^{\tilde{B}_k^\pi}\right\|_2 \Big| \tilde{S}_{k-N} = s\right] \mathbb{P}[\tilde{S}_{k-N} = s] \\
&\quad + \sum_{s \in \mathcal{S}} \mathbb{E}\left[\left\|D^\pi R^\pi - D^{\tilde{B}_k^\pi} R^{\tilde{B}_k^\pi}\right\|_2 \Big| \tilde{S}_{k-N} = s\right] \left(\mathbb{P}[S_{k-N} = s] - \mathbb{P}[\tilde{S}_{k-N} = s]\right) \\
&\leq |\mathcal{S}|^{\frac{5}{2}}|\mathcal{A}|R_{\max}\sqrt{\frac{t_2^{\mathrm{mix}}}{|\tilde{B}_k^\pi|}} + 2\sqrt{|\mathcal{S}|}R_{\max}d_{\mathrm{TV}}(\mu_{S_\infty}^\pi, \mu_{S_{k-N}}^\pi).
\end{aligned}
$$

This completes the proof. $\qquad\square$

### A.7 Proof of Lemma 3.3

*Proof.* First of all, the triangle inequality applied to the noise term in (13) leads to

$$
\mathbb{E}[\|w(M_k^\pi, V_k)\|_2] = \mathbb{E}\left[\left\|\frac{1}{|M_k^\pi|}\sum_{i=1}^{|M_k^\pi|} \delta(O_i^k; V_k) - \Delta_k(V_k) + \Delta_k(V_k) - \Delta_\pi(V_k)\right\|_2\right]
$$

$$\leq \mathbb{E}\left[\left\|\frac{1}{|M_k^\pi|}\sum_{i=1}^{|M_k^\pi|}\delta(O_i^k;V_k)-\Delta_k(V_k)\right\|_2\right]+\mathbb{E}\left[\|\Delta_k(V_k)-\Delta_\pi(V_k)\|_2\right]. \tag{18}$$

The first term in (18) is the difference of the empirically expected TD-errors in terms of replay buffer and mini-batch, which is bounded as

$$\mathbb{E}\left[\left\|\frac{1}{|M_k^\pi|}\sum_{i=1}^{|M_k^\pi|}\delta(O_i^k;V_k)-\Delta_k(V_k)\right\|_2\right]\leq\frac{2\sqrt{|\mathcal{S}|}R_{\max}}{1-\gamma}\left(\mathbb{E}\left[\left\|\frac{1}{|M_k^\pi|}\sum_{(s,r,s')\in M_k^\pi}(e_s e_s^\top - D^{B_k^\pi})\right\|_2\right]\right.$$

$$+\mathbb{E}\left[\left\|\frac{1}{|M_k^\pi|}\sum_{(s,s')\in M_k^\pi}\gamma(e_s e_{s'}^\top - D^{B_k^\pi}P^{B_k^\pi})\right\|_2\right]\right)$$

$$+\mathbb{E}\left[\left\|\frac{1}{|M_k^\pi|}\sum_{(s,r,s')\in M_k^\pi}(e_s r - D^{B_k^\pi}R^{B_k^\pi})\right\|_2\right], \tag{19}$$

where the inequality is obtained simply by expanding the terms in (11) and (5) and applying triangle inequality, together with the boundedness of $V_k$ and reward function in Assumption 2.4 and Lemma 3.1, respectively.

As the next step, we will upper bound the above terms using Bernstein inequality (Tropp et al., 2015). In particular, note that for $(s,r,s')\in M_k^\pi$, $s$ can be thought as a uniform sample from $D^{B_k^\pi}$. Hence, using the matrix concentration inequality in Lemma A.1 in Appendix, one gets

$$\mathbb{E}\left[\left\|\frac{1}{|M_k^\pi|}\sum_{(s,r,s')\in M_k^\pi}(e_s e_s^\top - D^{B_k^\pi})\right\|_2\right]\leq\sqrt{\frac{8\log(2|\mathcal{S}|)}{|M_k^\pi|}},$$

which is obtained by letting $\sigma=1$ and $X_{\max}=1$ in Lemma A.10 in the Appendix because $\|\mathbb{E}[e_s e_s^\top]\|_2\leq 1$ and $\|e_s e_s^\top\|_2\leq 1$.

Moreover, noting that $\|\mathbb{E}[e_s e_{s'}^\top]\|_2\leq\mathbb{E}[\|e_s e_{s'}^\top\|_2]\leq 1$, and $\|e_s e_{s'}^\top\|_2\leq 1$, we get the following bound on the second term:

$$\mathbb{E}\left[\left\|\frac{1}{|M_k^\pi|}\sum_{(s,r,s')\in M_k^\pi}\gamma(e_s e_{s'}^\top - D^{B_k^\pi}P^{B_k^\pi})\right\|_2\right]\leq 2\gamma\sqrt{\frac{2\log(2|\mathcal{S}|)}{|M_k^\pi|}}.$$

In a similar sense, the third term can be bounded as

$$\mathbb{E}\left[\left\|\frac{1}{|M_k^\pi|}\sum_{(s,r,s')\in M_k^\pi}(e_s r - D^{B_k^\pi}R^{B_k^\pi})\right\|_2\right]\leq 2R_{\max}\sqrt{\frac{2\log(2|\mathcal{S}|)}{|M_k^\pi|}},$$

which uses the inequalities, $\|\mathbb{E}[e_s r]\|_2\leq R_{\max}$ and $\|\mathbb{E}[r^2 e_s e_s^\top]\|_2\leq R_{\max}^2$. Collecting the above three inequalities yields the upper bound on (19), which is

$$\mathbb{E}\left[\left\|\frac{1}{|M_k^\pi|}\sum_{i=1}^{|M_k^\pi|}\delta(O_i^k;V_k)-\Delta_k(V_k)\right\|_2\right]\leq\frac{4\sqrt{|\mathcal{S}|}R_{\max}}{1-\gamma}\sqrt{\frac{8\log(2|\mathcal{S}|)}{|M_k^\pi|}}. \tag{20}$$

Now, we turn our attention to he second term, $\mathbb{E}\left[\|\Delta_\pi(V_k)-\Delta_k(V_k)\|_2\right]$, in (18), which is the difference between the expected TD-error with respect to the stationary distribution and the replay buffer distribution. Plugging the definitions in (11) and (12) into (18) yields

$$\mathbb{E}\left[\|\Delta_\pi(V_k)-\Delta_k(V_k)\|_2\right]$$

$$\leq \frac{2\sqrt{|\mathcal{S}|}R_{\max}}{1-\gamma}\mathbb{E}\left[\left(\left\|D^{\pi}-D^{B_k^{\pi}}\right\|_2+\left\|D^{\pi}P^{\pi}-D^{B_k^{\pi}}P^{B_k^{\pi}}\right\|_2\right)\right]+\mathbb{E}\left[\left\|D^{\pi}R^{\pi}-D^{B_k^{\pi}}R^{B_k^{\pi}}\right\|_2\right]$$

$$\leq \frac{2\sqrt{|\mathcal{S}|}R_{\max}}{1-\gamma}\left(3|\mathcal{S}|^2|\mathcal{A}|\sqrt{\frac{\tau^{\mathrm{mix}}}{|B_k^{\pi}|}}+16|\mathcal{S}|d_{\mathrm{TV}}(\mu_{S_{\infty}}^{\pi},\mu_{S_{k-N}}^{\pi})\right). \tag{21}$$

where the first inequality is due to the boundedness of $V_k$ in Lemma 3.1, and the last inequality follows from Lemma A.8. Collecting the terms in (20) and (21), we obtain the desired result.

$\square$

## A.8 Proof of Lemma 3.4 (Bound on second moment of the noise term)

*Proof.* Expanding the noise term in (9) we get

$$\mathbb{E}[||w(M_k^{\pi};V_k)||_2^2]\leq 2\left(\mathbb{E}\left[\left\|\frac{1}{L}\sum_{i=1}^{L}\delta(O_i^k;V_k)-\Delta_k(V_k)\right\|_2^2\right]+\mathbb{E}\left[\|\Delta_{\pi}(V_k)-\Delta_k(V_k)\|_2^2\right]\right), \tag{22}$$

where the inequality follows from the fact that $||a+b||^2\leq 2||a||^2+2||b||^2$ for $a,b\in\mathbb{R}^n$. In what follows, each terms in the above inequality will be bounded. First of all, the term $\mathbb{E}\left[\left\|\frac{1}{L}\sum_{i=1}^{L}\delta(O_i^k;V_k)-\Delta_k(V_k)\right\|_2^2\right]$ in (22) is bounded as

$$\mathbb{E}\left[\left\|\frac{1}{L}\sum_{i=1}^{L}\delta(O_i^k;V_k)-\Delta_k(V_k)\right\|_2^2\right]$$

$$=\mathbb{E}\left[\left\|\frac{1}{L}\sum_{(s,r,s')\in M_k}(e_sr-D^{B_k^{\pi}}R^{B_k^{\pi}})+(e_se_s^{\top}-D^{B_k})(V_k)+\gamma(e_se_{s'}^{\top}-D^{B_k}P^{B_k})V_k\right\|_2^2\right]$$

$$\leq\frac{3|\mathcal{S}|R_{\max}^2}{(1-\gamma)^2}\left(\mathbb{E}\left[\left\|\frac{1}{L}\sum_{(s,r,s')\in M_k}(e_se_s^{\top}-D^{B_k^{\pi}})\right\|_2^2\right]+\mathbb{E}\left[\left\|\frac{1}{L}\sum_{(s,r,s')\in M_k}\gamma(e_se_{s'}^{\top}-D^{B_k^{\pi}}P^{B_k^{\pi}})\right\|_2^2\right]\right)$$

$$+3\mathbb{E}\left[\left\|\frac{1}{L}\sum_{(s,r,s')\in M_k}(e_sr-D^{B_k^{\pi}}R^{B_k^{\pi}})\right\|_2^2\right]$$

$$\leq\frac{|\mathcal{S}|(1+R_{\max})^2}{(1-\gamma)^2}\frac{480}{|M_k^{\pi}|}(\log(2|\mathcal{S}|))^2,$$

where the first equality is obtained by expanding the TD-error term in (5) and expected TD-error in terms of replay buffer in (11). The first inequality follows from the fact that $||a+b+c||^2\leq 3(||a||^2+||b||^2+||c||^2)$ for $a,b,c\in\mathbb{R}^n$, and from the fact that the iterate $V_k$ is bounded in (3.1). We use concentration inequality in Corollary A.2 to bound the last inequality. To bound $\mathbb{E}\left[\left\|\frac{1}{L}\sum_{(s,r,s')\in M_k}(e_se_s^{\top}-D^{B_k^{\pi}})\right\|_2^2\right]$, we let $\sigma=1$ and $X_{\max}=1$ in Corollary A.2 since $||\mathbb{E}[e_se_s^{\top}]||_2\leq 1$ and $||e_se_s^{\top}||_2\leq 1$. Moreover, we can bound remaining terms in similar sense, which yields the last inequality.

To proceed further, Lemma A.9 is needed, which is provided after this proof. Applying Lemma A.9, the term $\mathbb{E}\left[\|\Delta_{\pi}(V_k)-\Delta_k(V_k)\|_2^2\right]$ is bounded as

$$\mathbb{E}\left[\|\Delta_{\pi}(V_k)-\Delta_k(V_k)\|_2^2\right]$$

$$=\mathbb{E}\left[\left\|(D^{\pi}-D^{B_k^{\pi}})V_k+(D^{\pi}R^{\pi}-D^{B_k^{\pi}}R^{B_k^{\pi}})+\gamma(D^{\pi}P^{\pi}-D^{B_k^{\pi}}P^{B_k^{\pi}})V_k\right\|_2^2\right]$$

$$\leq \frac{3|\mathcal{S}|R_{\max}^2}{(1-\gamma)^2}\left(4|\mathcal{S}|^4|\mathcal{A}|^2\frac{\max\{t_1^{\mathrm{mix}}, t_2^{\mathrm{mix}}\}}{|B_k^\pi|} + 8|\mathcal{S}|d_{\mathrm{TV}}(\mu_{S_\infty}^\pi, \mu_{S_{k-N}}^\pi)\right),$$

where the inequality follows from applying the bound on $V_k$ in Lemma 3.1, and collecting the terms in Lemma A.9. This completes the proof. $\qquad\square$

The following two lemmas have been used in the proof above, and they are formally introduced in the sequel.

**Lemma A.9.** *For $k \geq 0$, we have*

1) $\mathbb{E}[||D^\pi - D^{B_k^\pi}||_2^2] \leq |\mathcal{S}|\frac{t_1^{\mathrm{mix}}}{|B_k^\pi|} + 8d_{\mathrm{TV}}(\mu_{S_\infty}^\pi, \mu_{S_{k-N}}^\pi),$

2) $\mathbb{E}[||D^\pi P^\pi - D^{B_k^\pi}P^{B_k^\pi}||_2^2] \leq |\mathcal{S}|^2\frac{t_2^{\mathrm{mix}}}{|B_k^\pi|} + 2|\mathcal{S}|d_{\mathrm{TV}}(\mu_{S_\infty}^\pi, \mu_{S_{k-N}}^\pi),$

3) $\mathbb{E}\left[\left\|D^\pi R^\pi - D^{B_k^\pi}R^{B_k^\pi}\right\|_2^2\right] \leq |\mathcal{S}|^5|\mathcal{A}|^2 R_{\max}^2\frac{t_2^{\mathrm{mix}}}{|B_k^\pi|} + 2|\mathcal{S}|R_{\max}^2 d_{\mathrm{TV}}(\mu_{S_\infty}^\pi, \mu_{S_{k-N}}^\pi).$

*Proof.* Let $\{\tilde{S}_k\}_{k \geq -N}$ be the stationary Markov chain defined in Section 3.3. Then, for the first statement, we get the following bounds:

$$\begin{aligned}\mathbb{E}[||D^\pi - D^{B_k^\pi}||_2^2] &= \sum_{s \in \mathcal{S}} \mathbb{E}\left[||D^\pi - D^{\tilde{B}_k^\pi}||_2^2 \Big| \tilde{S}_{k-N} = s\right] \mathbb{P}[S_{k-N} = s]\\ &= \sum_{s \in \mathcal{S}} \mathbb{E}[||D^\pi - D^{\tilde{B}_k^\pi}||_2^2 \mid \tilde{S}_{k-N} = s]\mathbb{P}[\tilde{S}_{k-N} = s]\\ &\quad + \sum_{s \in \mathcal{S}} \mathbb{E}[||D^\pi - D^{\tilde{B}_k^\pi}||_2^2 \mid \tilde{S}_{k-N} = s](\mathbb{P}[S_{k-N} = s] - \mathbb{P}[\tilde{S}_{k-N} = s])\\ &\leq |\mathcal{S}|\frac{t_1^{\mathrm{mix}}}{|B_k^\pi|} + 8d_{\mathrm{TV}}(\mu_{S_\infty}^\pi, \mu_{S_{k-N}}^\pi),\end{aligned}$$

where the first equality is due to the law of total expectation, and the fact that $\mathbb{P}[S_{k-N+1}, S_{k-N+2}, \ldots, S_k \mid S_{k-N}]$ and $\mathbb{P}[\tilde{S}_{k-N+1}, \tilde{S}_{k-N+2}, \ldots, \tilde{S}_k \mid \tilde{S}_{k-N}]$ are identical. Moreover, the second equality follows from simple algebraic decomposition, and the inequality is due to the fact that $||A - B||_2^2 \leq 2(||A||_2^2 + ||B||_2^2)$ for $A, B \in \mathbb{R}^{|\mathcal{S}| \times |\mathcal{S}|}$, and applies Lemma A.10 and the definition of total variation distance in Definition 2.1. Note that Lemma A.10 will be given after the proof.

For the second statement, it follows that

$$\begin{aligned}\mathbb{E}\left[\left\|D^\pi P^\pi - D^{B_k}P^{B_k}\right\|_2^2\right] &= \sum_{s \in \mathcal{S}} \mathbb{E}\left[\left\|D^\pi P^\pi - D^{B_k}P^{B_k}\right\|_2^2 \mid S_{k-N} = s\right]\mathbb{P}[S_{k-N} = s]\\ &\leq \mathbb{E}\left[\left\|D^\pi P^\pi - D^{\tilde{B}_k^\pi}P^{\tilde{B}_k^\pi}\right\|_2^2\right] + 2|\mathcal{S}|d_{\mathrm{TV}}(\mu_{S_\infty}^\pi, \mu_{S_{k-N}}^\pi)\\ &\leq |\mathcal{S}|^2\frac{t_2^{\mathrm{mix}}}{|\tilde{B}_k^\pi|} + 2|\mathcal{S}|d_{\mathrm{TV}}(\mu_{S_\infty}^\pi, \mu_{S_{k-N}}^\pi),\end{aligned}$$

where the first inequality is due to the fact that $||P^\pi||_2^2 \leq |\mathcal{S}|$, $||P^{\tilde{B}_k^\pi}||_2^2 \leq |\mathcal{S}|$ since $P^\pi$ and $P^{\tilde{B}_k^\pi}$ are stochastic matrix, i.e., the row sum equals one. This concludes the proof of the second statement.

Next, similar arguments hold for $\mathbb{E}\left[\left\|D^\pi R^\pi - D^{B_k^\pi}R^{B_k^\pi}\right\|_2^2\right]$ as follows:

$$\begin{aligned}\mathbb{E}\left[\left\|D^\pi R^\pi - D^{B_k^\pi}R^{B_k^\pi}\right\|_2^2\right] &= \sum_{s \in \mathcal{S}} \mathbb{E}\left[\left\|D^\pi R^\pi - D^{\tilde{B}_k^\pi}R^{\tilde{B}_k^\pi}\right\|_2^2 \Big| S_{k-N} = s\right]\mathbb{P}[S_{k-N} = s]\\ &\leq \mathbb{E}\left[\left\|D^\pi R^\pi - D^{\tilde{B}_k^\pi}R^{\tilde{B}_k^\pi}\right\|_2^2\right] + 2|\mathcal{S}|R_{\max}^2 d_{\mathrm{TV}}(\mu_{S_\infty}^\pi, \mu_{S_{k-N}}^\pi)\\ &\leq |\mathcal{S}|^5|\mathcal{A}|^2 R_{\max}^2\frac{t_2^{\mathrm{mix}}}{|B_k^\pi|} + 2|\mathcal{S}|R_{\max}^2 d_{\mathrm{TV}}(\mu_{S_\infty}^\pi, \mu_{S_{k-N}}^\pi).\end{aligned}$$

This completes the proof. □

**Lemma A.10.** *For $k \geq 0$, we have*

*1)* $\mathbb{E}\left[\left\|D^\pi - D^{\tilde{B}_k^\pi}\right\|_2^2\right] \leq |\mathcal{S}| \frac{t_1^{\mathrm{mix}}}{|\tilde{B}_k^\pi|}$,

*2)* $\mathbb{E}\left[\left\|D^\pi P^\pi - D^{\tilde{B}_k^\pi} P^{\tilde{B}_k^\pi}\right\|_2^2\right] \leq |\mathcal{S}|^2 \frac{t_2^{\mathrm{mix}}}{|\tilde{B}_k^\pi|}$,

*3)* $\mathbb{E}\left[\left\|D^\pi R^\pi - D^{\tilde{B}_k^\pi} R^{\tilde{B}_k^\pi}\right\|_2^2\right] \leq |\mathcal{S}|^5 |\mathcal{A}|^2 R_{\max}^2 \frac{t_2^{\mathrm{mix}}}{|\tilde{B}_k^\pi|}$.

*Proof.* The first statement is proved via the following inequalities:

$$\mathbb{E}\left[\left\|D^\pi - D^{\tilde{B}_k^\pi}\right\|_2^2\right] \leq \sum_{s \in \mathcal{S}} \mathbb{E}\left[\left(\mu_{S_\infty}^\pi(s) - \mu_{\tilde{S}}^{\tilde{B}_k^\pi}(s)\right)^2\right] \leq |\mathcal{S}| \frac{t_1^{\mathrm{mix}}}{|\tilde{B}_k^\pi|}.$$

Here, the first inequality follows from the matrix norm inequality $||\cdot||_2 \leq ||\cdot||_F$, where $||\cdot||_F$ stands for Frobenius norm, and the second inequality applies Lemma A.3. The proof of the second statement follows similar lines. In particular, letting $\mu_{\tilde{S},\tilde{S}'}^{\tilde{B}_k^\pi}$ stand for stationary distribution of $\{(\tilde{S}_k, \tilde{S}_{k+1})\}_{k \geq -N}$, one gets

$$\mathbb{E}\left[\left\|D^\pi P^\pi - D^{\tilde{B}_k^\pi} P^{\tilde{B}_k^\pi}\right\|_2^2\right] \leq \sum_{(s,r,s') \in \mathcal{O}} \mathbb{E}\left[(\mu_{S_\infty,S_\infty'}^\pi(s,s') - \mu_{\tilde{S},\tilde{S}'}^{\tilde{B}_k^\pi}(s,s'))^2\right] \leq |\mathcal{S}|^2 \frac{t_2^{\mathrm{mix}}}{|\tilde{B}_k^\pi|},$$

where the first inequality follows from the relation between spectral norm and Frobenius norm, and the second inequality applies Lemma A.3.

For the third statement, one can bound $\mathbb{E}\left[\left\|D^\pi R^\pi - D^{\tilde{B}_k^\pi} R^{\tilde{B}_k^\pi}\right\|_2^2\right]$ with the same logic as in the proof of Lemma 3.3 as follows:

$$\mathbb{E}\left[\left\|D^\pi R^\pi - D^{\tilde{B}_k^\pi} R^{\tilde{B}_k^\pi}\right\|_2^2\right] = \mathbb{E}\left[\sum_{s \in \mathcal{S}}\left(\mathbb{E}\left[r(S,A,S') \mid S=s, \pi\right] \mu_{S_\infty}^\pi(s) - \mu_{\tilde{S}}^{\tilde{B}_k^\pi}(s)[R^{\tilde{B}_k^\pi}]_s\right)^2\right]$$

$$\leq |\mathcal{S}|^3 |\mathcal{A}|^2 R_{\max}^2 \mathbb{E}\left[\sum_{s \in \mathcal{S}} \sum_{s' \in \mathcal{S}}\left(\mu_{S_\infty,S_\infty'}^\pi(s,s') - \mu_{\tilde{S},\tilde{S}'}^{\tilde{B}_k^\pi}(s,s')\right)^2\right]$$

$$\leq |\mathcal{S}|^5 |\mathcal{A}|^2 R_{\max}^2 \frac{t_2^{\mathrm{mix}}}{|\tilde{B}_k^\pi|}.$$

This completes the proof. □

## A.9 Proof of Theorem 3.5

*Proof.* We prove the first statement. Using (7), $||V_{k+1} - V^\pi||_M^2$ can be expanded as follows:

$$||V_{k+1} - V^\pi||_M^2$$
$$= (V_k - V^\pi)^\top A^\top M A (V_k - V^\pi) + 2\alpha (V_k - V^\pi)^\top A^\top M w(M_k^\pi, V_k) + \alpha^2 ||w(M_k^\pi, V_k)||_M^2$$
$$= ||V_k - V^\pi||_M^2 - ||V_k - V^\pi||_2^2 + 2\alpha (V_k - V^\pi)^\top A^\top M w(M_k^\pi, V_k) + \alpha^2 ||w(M_k^\pi, V_k)||_M^2,$$

where the first equality follows from (7), and the last equality uses the Lyapunov equation in (10). Taking expectation yields

$$\mathbb{E}[||V_{k+1} - V^\pi||_M^2]$$
$$= \mathbb{E}[||V_k - V^\pi||_M^2] - \mathbb{E}[||V_k - V^\pi||_2] + 2\alpha \mathbb{E}[(V_k - V^\pi)^\top A^\top M w(M_k^\pi, V_k)] + \alpha^2 \mathbb{E}[||w(M_k^\pi, V_k)||_M^2]$$

$$
\leq \mathbb{E}[||V_k - V^\pi||_M^2] - \mathbb{E}[||V_k - V^\pi||_2] + \frac{8|\mathcal{S}|^{\frac{3}{2}}R_{\max}}{(1-\gamma)^2\mu_{\min}^\pi}\mathbb{E}[||w(M_k^\pi, V_k)||_2]
$$

$$
+ \frac{2\alpha|\mathcal{S}|}{(1-\gamma)\mu_{\min}^\pi}\mathbb{E}[||w(M_k^\pi, V_k)||_2^2]
$$

$$
\leq \mathbb{E}\left[||V_k - V^\pi||_M^2\right] - \mathbb{E}[||V_k - V^\pi||_2^2]
$$

$$
+ \frac{32|\mathcal{S}|^2 R_{\max}^2}{(1-\gamma)^3\mu_{\min}^\pi}\left(\sqrt{\frac{8\log(2|\mathcal{S}|)}{L}} + 2|\mathcal{S}|^{\frac{3}{2}}|\mathcal{A}|\sqrt{\frac{\tau^{\mathrm{mix}}}{N}} + 16|\mathcal{S}|d_{\mathrm{TV}}(\mu_{S_\infty}^\pi, \mu_{S_{k-N}}^\pi)\right)
$$

$$
+ \alpha\frac{4|\mathcal{S}|^2(R_{\max}+1)^2}{(1-\gamma)^3\mu_{\min}^\pi}\left(\frac{120(\log(2|\mathcal{S}|))^2}{L} + 4|\mathcal{S}|^4|\mathcal{A}|^2\frac{\tau^{\mathrm{mix}}}{N} + 8|\mathcal{S}|d_{\mathrm{TV}}(\mu_{S_\infty}^\pi, \mu_{S_{k-N}}^\pi)\right),
$$

where the first inequality follows from Cauchy-Schwartz inequality, and the boundedness of $V_k$ in Lemma 3.1. The last inequality comes from bounding the first and second moment of $||w(M_k^\pi, V_k)||_2$ by Lemma 3.3 and Lemma 3.4, respectively.

Summing the last inequality from $k = 0$ to $k = T - 1$ and dividing both sides by $T$, one gets

$$
\frac{1}{T}\sum_{k=0}^{T-1}\mathbb{E}[||V_k - V^\pi||_2^2] \leq \frac{1}{T}\mathbb{E}[||V_0 - V^\pi||_M^2]
$$

$$
+ \frac{32|\mathcal{S}|^2 R_{\max}^2}{(1-\gamma)^3\mu_{\min}^\pi}\left(\sqrt{\frac{8\log(2|\mathcal{S}|)}{L}} + 2|\mathcal{S}|^{\frac{3}{2}}|\mathcal{A}|\sqrt{\frac{\tau^{\mathrm{mix}}}{N}}\right)
$$

$$
+ \frac{1}{T}\frac{32|\mathcal{S}|^2 R_{\max}^2}{(1-\gamma)^3\mu_{\min}^\pi}\left(\sum_{k=0}^{T-1}16|\mathcal{S}|d_{\mathrm{TV}}(\mu_{S_\infty}^\pi, \mu_{S_{k-N}}^\pi)\right)
$$

$$
+ \alpha\frac{4|\mathcal{S}|^2(R_{\max}+1)^2}{(1-\gamma)^3\mu_{\min}^\pi}\left(\frac{120(\log(2|\mathcal{S}|))^2}{L} + 4|\mathcal{S}|^4|\mathcal{A}|^2\frac{\tau^{\mathrm{mix}}}{N}\right)
$$

$$
+ \alpha\frac{1}{T}\frac{4|\mathcal{S}|^2 R_{\max}^2}{(1-\gamma)^3\mu_{\min}^\pi}\left(\sum_{k=0}^{T-1}16|\mathcal{S}|d_{\mathrm{TV}}(\mu_{S_\infty}^\pi, \mu_{S_{k-N}}^\pi)\right).
$$

To bound the sum of total variation terms in the above inequality, the following relation can be used:

$$
\sum_{k=0}^{T-1}d_{\mathrm{TV}}(\mu_{S_\infty}^\pi, \mu_{S_{k-N}}^\pi) \leq t_1^{\mathrm{mix}}\sum_{k=0}^{\lfloor T/t_1^{\mathrm{mix}}\rfloor}d_{\mathrm{TV}}(\mu_{S_\infty}^\pi, \mu_{S_{kt_1^{\mathrm{mix}}-N}}^\pi) \leq t_1^{\mathrm{mix}}\sum_{k=0}^{\lfloor T/t_1^{\mathrm{mix}}\rfloor}2^{-k} \leq 2t_1^{\mathrm{mix}},
$$

where the first inequality follows from non-increasing property of total variation of irreducible and aperiodic Markov chain in Lemma A.4 in Appendix. Moreover, the second inequality follows from the first item in Lemma A.4 in Appendix. Combining the last two inequalities leads to

$$
\frac{1}{T}\sum_{k=0}^{T-1}\mathbb{E}[||V_k - V^\pi||_2^2]
$$

$$
\leq \frac{1}{T}\frac{2|\mathcal{S}|}{\alpha(1-\gamma)}\mathbb{E}[||V_0 - V^\pi||_2^2]
$$

$$
+ \frac{32|\mathcal{S}|^2 R_{\max}^2}{(1-\gamma)^3\mu_{\min}^\pi}\left(\sqrt{\frac{\log(8|\mathcal{S}|)}{L}} + 2|\mathcal{S}|^{\frac{3}{2}}|\mathcal{A}|\sqrt{\frac{\tau^{\mathrm{mix}}}{N}}\right) + \frac{32^2|\mathcal{S}|^2 R_{\max}^2}{(1-\gamma)^3\mu_{\min}^\pi}\frac{2t_1^{\mathrm{mix}}}{T}
$$

$$
+ \alpha\frac{4|\mathcal{S}|^2(R_{\max}+1)^2}{(1-\gamma)^3\mu_{\min}^\pi}\left(\frac{120(\log(2|\mathcal{S}|))^2}{L} + 4|\mathcal{S}|^4|\mathcal{A}|^2\frac{\tau^{\mathrm{mix}}}{N}\right) + \alpha\frac{128|\mathcal{S}|^2 R_{\max}^2}{(1-\gamma)^3\mu_{\min}^\pi}\frac{t_1^{\mathrm{mix}}}{T}.
$$

This completes the proof of first statement. The second statement can be directly obtained from applying Jensen's inequality.

$\square$

### A.10 Proof of Theorem 3.6

*Proof.* First of all, we prove the first statement. The term $\|V_{k+1} - V^\pi\|_2^2$ is expanded according to the update in (7) as follows:

$$
\|V_{k+1} - V^\pi\|_2^2
$$
$$
=\|V_k - V^\pi\|_{A^\top A}^2 + 2\alpha(V_k - V^\pi)A^\top w(M_k^\pi, V_k) + \alpha^2\|w(M_k^\pi, V_k)\|_2^2
$$
$$
=(V_0 - V^\pi)^\top (A^\top)^{k+1}A^{k+1}(V_0 - V^\pi) + \sum_{i=0}^{k} 2\alpha(V_i - V^\pi)^\top A^\top (A^\top)^{k-i}A^{k-i}w(M_i^\pi, V_i) \tag{23}
$$
$$
+ \alpha^2 \sum_{i=0}^{k} w(M_i^\pi; V_i)^\top (A^\top)^{k-i}A^{k-i}w(M_i^\pi; V_i), \tag{24}
$$

where the second equality follows from the fact that for $0 \le i \le k-1$, the $i$-th expansion results to weighted inner product of $A(V_{k-i-1} - V^\pi)$ and $w(M_{k-i-1}^\pi, V_{k-i-1})$ and weighted squared norm of $w(M_{k-i-1}^\pi, V_{k-i-1})$ which are (23) and (24), respectively.

Next, applying Cauchy–Schwartz inequality to (24) yields

$$
\|V_{k+1} - V^\pi\|_2^2
$$
$$
\le \|V_0 - V^\pi\|_2^2 |\mathcal{S}|\|A^{2k+2}\|_\infty + 2\alpha|\mathcal{S}| \sum_{i=0}^{k} \|V_i - V^\pi\|_2 \|A^{k-i+1}\|_\infty \|A^{k-i}\|_\infty \|w(M_i^\pi; V_i)\|_2
$$
$$
+ \alpha^2 \sum_{i=0}^{k} \|A^{2k-2i}\|_2 \|w(M_i^\pi; V_i)\|_2^2
$$
$$
\le \|V_0 - V^\pi\|_2^2 |\mathcal{S}|\|A\|_\infty^{2k+2} + 2\alpha|\mathcal{S}| \sum_{i=0}^{k} \|V_i - V^\pi\|_2 \|A\|_\infty^{2k-2i+1} \|w(M_i^\pi; V_i)\|_2
$$
$$
+ \alpha^2 \sqrt{|\mathcal{S}|} \sum_{i=0}^{k} \|A\|_\infty^{2k-2i} \|w(M_i^\pi; V_i)\|_2^2,
$$

where the first inequality follows from the matrix norm inequality $\|A\|_2 \le \sqrt{|\mathcal{S}|}\|A\|_\infty$, and the last inequality is due to sub-multiplicativity of the matrix norm. After taking expectation, we get

$$
\mathbb{E}\left[\|V_{k+1} - V^\pi\|_2^2\right]
$$
$$
\le \|V_0 - V^\pi\|_2^2 \|\mathcal{S}\|\|A^{k+1}\|_2^2
$$
$$
+ \alpha \frac{64|\mathcal{S}|^2 R_{\max}^2}{(1-\gamma)^2} \left(\sqrt{\frac{8\log(2|\mathcal{S}|)}{L}} + |\mathcal{S}|^{\frac{3}{2}}|\mathcal{A}|\sqrt{\frac{\tau^{\mathrm{mix}}}{N}}\right) \sum_{i=0}^{k} \|A\|_\infty^{2k-2i+1}
$$
$$
+ \alpha^2 \sqrt{|\mathcal{S}|} \frac{4|\mathcal{S}|(R_{\max} + 1)^2}{(1-\gamma)^2} \left(\frac{120(\log(2|\mathcal{S}|)^2}{|M_k^\pi|} + 4|\mathcal{S}|^4|\mathcal{A}|^2 \frac{\tau^{\mathrm{mix}}}{|B_k^\pi|}\right) \sum_{i=0}^{k} \|A\|_\infty^{2k-2i}
$$
$$
\le \|V_0 - V^\pi\|_2^2 \|\mathcal{S}\|(1 - \alpha(1-\gamma)\mu_{\min}^\pi)^{2k+2}
$$
$$
+ \frac{64|\mathcal{S}|^2 R_{\max}^2}{(1-\gamma)^3 \mu_{\min}^\pi} \left(\sqrt{\frac{8\log(2|\mathcal{S}|)}{L}} + |\mathcal{S}|^{\frac{3}{2}}|\mathcal{A}|\sqrt{\frac{\tau^{\mathrm{mix}}}{N}}\right)
$$
$$
+ \alpha \frac{4|\mathcal{S}|(R_{\max} + 1)^2}{(1-\gamma)^3 \mu_{\min}^\pi} \left(\frac{120(\log(2|\mathcal{S}|)^2}{|M_k^\pi|} + 4|\mathcal{S}|^4|\mathcal{A}|^2 \frac{\tau^{\mathrm{mix}}}{|B_k^\pi|}\right),
$$

where the first inequality follows from Lemma 3.3, 3.4, and the fact that $d_{\mathrm{TV}}(\mu_{S_\infty}^\pi, \mu_{S_{k-N}}^\pi)$ is zero for all $k \ge 0$ because the initial state distribution is identical to the stationary distribution. Moreover, the second inequality comes from the bound on $\|A\|_\infty$ in Lemma 3.2. This proves the first statement. Finally, applying Jensen's inequality to the first item which is the bound on $\mathbb{E}\left[\|V_k - V^\pi\|_2^2\right]$, leads to the second statement. $\square$

### A.11 Sample complexity result of Theorem 3.6

The result of Bhandari et al. (2018) requires following number of samples to bound (17) with $\epsilon > 0$:

$$\tilde{\mathcal{O}}\left(\frac{1}{\epsilon^2}\frac{t^{\mathrm{mix}}}{(1-\gamma)^3(\mu^\pi_{\min})^2}\right), \tag{25}$$

where we used $t^{\mathrm{mix}}(\alpha) = \tilde{\mathcal{O}}\left(t^{\mathrm{mix}}\right)$ from Lemma A.5 in the Appendix, and $t^{\mathrm{mix}}(\alpha)$ is defined in (1).

To bound the second line of the the second item in Theorem 3.6 with $\epsilon$, we require

$$L = \tilde{\mathcal{O}}\left(\frac{|\mathcal{S}|^4}{(1-\gamma)^6(\mu^\pi_{\min})^4}\frac{1}{\epsilon^4}\right), \quad N = \tilde{\mathcal{O}}\left(\frac{\tau_{\mathrm{mix}}|\mathcal{S}|^7|\mathcal{A}|^2}{(1-\gamma)^6(\mu^\pi_{\min})^4}\frac{1}{\epsilon^4}\right).$$

Next, to bound the third line of the the second item in Theorem 3.6 with $\epsilon$, we require

$$L = \tilde{\mathcal{O}}\left(\frac{|\mathcal{S}|}{(1-\gamma)^3\mu^\pi_{\min}}\frac{1}{\alpha}\frac{1}{\epsilon^2}\right), \quad N = \tilde{\mathcal{O}}\left(\frac{|\mathcal{S}|^5|\mathcal{A}|^2\tau^{\mathrm{mix}}}{(1-\gamma)^3\mu^\pi_{\min}}\frac{1}{\alpha}\frac{1}{\epsilon^2}\right).$$

Lastly, to bound the first line of the the second item in Theorem 3.6 with $\epsilon$, the number of steps $k$ needs to satisfy the following:

$$k = \mathcal{O}\left(\frac{1}{\alpha(1-\gamma)\mu^\pi_{\min}}\ln\left(\frac{1}{\epsilon}\right)\right).$$

The overall sample complexity results to

$$k + N = \tilde{\mathcal{O}}\left(\frac{|\mathcal{S}|^7|\mathcal{A}|^2\tau^{\mathrm{mix}}}{(1-\gamma)^6(\mu^\pi_{\min})^4}\frac{1}{\alpha}\frac{1}{\epsilon^4}\right).$$

Compared to the sample complexity in (25) by Bhandari et al. (2018), the above sample complexity result leaves room for improvement in terms of $\frac{1}{1-\gamma}$ and $\frac{1}{\mu^\pi_{\min}}$. However, if we consider, $N$, as given pre-collected dataset, the sample complexity result is only $\mathcal{O}\left(\frac{1}{\alpha(1-\gamma)\mu^\pi_{\min}}\ln\left(\frac{1}{\epsilon}\right)\right)$, where $\alpha \in (0,1)$.

