# OpenReview forum: "Finite-Time Analysis of Temporal Difference Learning with Experience Replay"
_TMLR — Accepted by TMLR_

### Review · Reviewer_EqyT · 2024-04-12

**Summary Of Contributions:**

The paper provides a finite-time analysis of on-policy tabular temporal difference learning with experience replay. It then provides a comparative analysis with existing results in the literature. While the convergence rate is on a similar order, they emphasize that the error induced by using a constant step-size can be controlled by the mini-batch and replay buffer sizes.

**Audience:**

Yes

**Broader Impact Concerns:**

N/A.

**Claims And Evidence:**

Yes

**Requested Changes:**

1. Can the authors acknowledge or clarify the impact of using the Markovian observation model on the results? It's unclear in the comparative analysis as the table does not present any other works with Experience replay (or comparable mini-batch updates). The intuitions for why the constant error term can be controlled by the mini-batch size seems like it should similarly apply to an on-policy tabular analog of Fitted Q-Iteration. As this seems to be the main result being emphasized by the paper, it would be good to detail how the paper's chosen setting may have played a role in the result, if at all.

2. The signs in Equation 7 seem incorrect--- Specifically, the expected TD-error with respect to the stationary distribution appears incorrect as the immediate reward and the discounted future value have opposing signs. The subsequent equations based on this appear correct, so this did not affect downstream results.

3. In Appendix A.3, the proof of Lemma 2.3 is given in-line and is extremely dense. Is there any particular reason for this? It seems out of place given the presentation of the other proofs in the paper, and is very hard to read. I think its presentation would benefit from being unrolled.

4. In Equation 3, the notation "e" is introduced with no definition or acknowledgement of it in any of the text preceding it. Later equations continue to use this undefined notation. I think it would improve clarity to explain this quantity, even informally.

**Strengths And Weaknesses:**

**Strengths:**
* The paper's organization was largely easy to follow.
* As far as I was able to check, the results appear correct.

**Weaknesses:**
* While the emphasized result of the constant error being largely independent from the choice of step size is interesting, it seems intuitive, especially in the tabular setting. The step-size is related to the window size of a moving average, where a sufficiently small step-size is often needed to average out any stochasticity in the update targets. In the experience replay update, we see that a sample average is computed over the mini-batch, alleviating the need for the step-size to perform this averaging.
* My primary concern with the paper is that they chose a very specific setup for their analysis, that it's difficult to situate the result among the space of similar setups. For example, while they emphasize that prior analysis on Fitted Q-Iteration had a very strong assumption on how transitions are sampled, it's unclear how the contrasting choice of a Markovian observation model manifests in the main result and the conclusions drawn. The emphasized result around the error term seems like it results from the use of mini-batch updates, and not be particularly unique to the chosen setup.

---

> ### Author Response · Authors · 2024-05-27
> **Reply to reviewer (Part 1)**
>
> **P1.** *While the emphasized result of the constant error being largely independent from the choice of step size is interesting, it seems intuitive, especially in the tabular setting. The step-size is related to the window size of a moving average, where a sufficiently small step-size is often needed to average out any stochasticity in the update targets. In the experience replay update, we see that a sample average is computed over the mini-batch, alleviating the need for the step-size to perform this averaging.*
>
> **A1.** Thank you for the valuable feedback. As the reviewer mentioned, even though the result can be intuitive, the proposed work is the first to rigorously analyze the relation between the error bound and mini-batch size, $L$, and replay buffer size, $N$, under the TD-learning setup. In this respect, we believe that the proposed work can be seen as a meaningful step toward deeper understanding of the replay buffer technique. Thank you for the attention on our manuscript.
>
> **P2.** *My primary concern with the paper is that they chose a very specific setup for their analysis, that it's difficult to situate the result among the space of similar setups. For example, while they emphasize that prior analysis on Fitted Q-Iteration had a very strong assumption on how transitions are sampled, it's unclear how the contrasting choice of a Markovian observation model manifests in the main result and the conclusions drawn. The emphasized result around the error term seems like it results from the use of mini-batch updates, and not be particularly unique to the chosen setup.*
>
> **A2.** Thank you for the valuable comment. First of all, we would like to note that the Markovian observation model is a more natural and realistic scenario than the i.i.d. observation model, which improves practicality of the analysis. The fitted Q-learning in [1] indeed does not take into account the replay buffer technique because they use i.i.d. samples, which is more close to the batch Q-learning with i.i.d. samples. The replay buffer technique is quite different from the batch Q-learning with i.i.d. transition samples because in the mini-batch approach, the samples are from the single episode trajectory in a moving window, which are highly correlated, and the correlation increases proportionally to the size of the window.
> As for the theoretical analysis, we would like to note that the analysis of Markovain observation model substantially differs from that of i.i.d. observation model due to correlation between the samples and the iterate. As a result, Markovian observation model incurs addition mixing time factor $\tau^{\text{mix}}$ in the final error bound. Next, we highlight the difference with prior analysis [1] on fitted Q-iteration. **1)** [1] focused on i.i.d. observation model, which is sampled from a fixed distribution that satisfies particular condition such that it is similar to the true underlying distribution over the state-action space. The assumption may not hold in our setting. We do not pose any assumptions and consider a time-varying experience replay buffer, i.e., a first-in-first-out queue, which is closer to the practice; **2)** The fitted Q-iteration algorithm considered in [1] is clearly different from TD-learning. The fitted Q-iteration algorithm is similar to stochastic gradient descent algorithm in supervised-learning problem, whereas TD-learning is an online learning algorithm which does not use any (stochastic) gradient of an objective function. This difference results in substantially different analysis steps. The related discussions have been newly added in the revised manuscript on page 2. We thank the reviewer for the constructive comments.

---

> > ### Author Response · Authors · 2024-05-27
> > **Reply to reviewer (Part 2)**
> >
> > **P3.** *Can the authors acknowledge or clarify the impact of using the Markovian observation model on the results? It's unclear in the comparative analysis as the table does not present any other works with Experience replay (or comparable mini-batch updates). The intuitions for why the constant error term can be controlled by the mini-batch size seems like it should similarly apply to an on-policy tabular analog of Fitted Q-Iteration. As this seems to be the main result being emphasized by the paper, it would be good to detail how the paper's chosen setting may have played a role in the result, if at all.*
> >
> > **A3.** Thank you for the valuable feedback. The main impact of using the Markovian observation model is the improved practicality of the analysis compared to the i.i.d. case. In particular, the Markovian observation model allows us to obtain transitions from a single trajectory of an episode which is more realistic and practical assumption. This setting has been also considered in numerous papers with TD-learning and Q-learning, while has not been addressed with the replay buffer scenario. Moreover, we would like to stress that TD-learning analysis with replay buffer has not been theoretically studied until now in the literature. As for fitted Q-learning in [1], it does not provide error bounds similar to the proposed one with the dependency on the memory size. For more detailed answers, please refer to A2. We have provided a detailed response on the issue. Thanks to the reviewer's valuable comments, the related discussions have also been included in the revision, which improves the quality of the paper.
> >
> > **P4.** *The signs in Equation 7 seem incorrect--- Specifically, the expected TD-error with respect to the stationary distribution appears incorrect as the immediate reward and the discounted future value have opposing signs. The subsequent equations based on this appear correct, so this did not affect downstream results.*
> >
> > **A4.** We deeply appreciate your valuable feedback and careful examinations of the manuscript. The errors indicated by the reviewer have been corrected in the revision.
> >
> > **P5.** *In Appendix A.3, the proof of Lemma 2.3 is given in-line and is extremely dense. Is there any particular reason for this? It seems out of place given the presentation of the other proofs in the paper, and is very hard to read. I think its presentation would benefit from being unrolled.*
> >
> > **A5.** We fully agree with the reviewer's opinion. The presentations are dense due to the page limits. Following the reviewer's comment, the dense presentations have been unrolled after simplifying the proof in the revision. Thank you for the attention on our paper.
> >
> > **P6.** *In Equation 3, the notation "e" is introduced with no definition or acknowledgement of it in any of the text preceding it. Later equations continue to use this undefined notation. I think it would improve clarity to explain this quantity, even informally.*
> >
> > **A6.** Thank you for the careful examination of the manuscript. Following the reviewer's comment, we have newly added its definition in the revised manuscript on page 5.
> >
> >
> > **References**
> >
> > [1] Fan, Jianqing, et al. "A theoretical analysis of deep Q-learning." Learning for dynamics and control. PMLR, 2020.

---

### Review · Reviewer_gLw3 · 2024-05-13

**Summary Of Contributions:**

This paper studies the convergence of TD-learning with experience replay. The authors establish the convergence of this algorithm based on the step size, and the replay buffer’s size and the mini-batch’s size. The authors argue that compared to previous work, their algorithm has the advantage of not having strict problem dependent requirements on the step size.

**Audience:**

Yes

**Broader Impact Concerns:**

No concern

**Claims And Evidence:**

Yes

**Requested Changes:**

- In equation (3), you did not define what e_s_k is.
Overall, besides comparing the requirements on the step size, can you also discuss about the sample complexity of TD-learning with experience replay, and under similar conditions (and assumptions), can you compare its sample complexity with normal TD-learning? that way we can have a clear view on potential advantages of this algorithm.

**Strengths And Weaknesses:**

- The authors claim that in their result the step size can be any constant in (0,1), and it does not need to be dependent on the horizon, and they count this as an advantage compared to the prior work. However, in this work we need to set L and N based on the horizon T prior to running the algorithm, which plays the same role as setting the step size depending on the horizon T. Can you elaborate more why setting L and N is better than setting the step size alpha?
- Furthermore, the comment about Lakshminarayanan & Szepesvari (2018) is not clear to me. Clearly that paper studies a more general setting with linear function approximation, and hence the step size requires knowledge of the problem structure, including the feature space.
- In this paper, rather than knowing the problem structure, we need the knowledge of the mixing time to set N>tau_mix. How is it different than the prior work?
- Similarly, in Theorem 3.6 you impose the strong assumption of S_{-N}\sim \mu_S^\pi. How is this assumption realistic, given that we do not know the stationary distribution of the underlying Markov chain?

---

> ### Author Response · Authors · 2024-05-27
> **Reply to reviewer (Part 1)**
>
> **P1.** *The authors claim that in their result the step size can be any constant in (0,1), and it does not need to be dependent on the horizon, and they count this as an advantage compared to the prior work. However, in this work we need to set L and N based on the horizon T prior to running the algorithm, which plays the same role as setting the step size depending on the horizon T. Can you elaborate more why setting L and N is better than setting the step size alpha?*
>
> **A1.** Thank you for the valuable comment, and we agree with the reviewer's comment. However, in the following, we would like to explain the case when setting $L$ and $N$ could be better than setting the step-size. **1)** Suppose that we are already given a sufficient amount of data as in the offline RL setting in [2], and then interaction with the environment is allowed. Then, the convergence rate, $\mathcal{O}(\exp(-\alpha k))$, becomes faster as we choose large $\alpha$, which is possible thanks to $\alpha \in (0,1)$. In other words, if we exclude $N$ in the sample complexity, permitting large $\alpha$ incurs low sample complexity result. However, if we take into account $N$ as a sample complexity result, the overall sample complexity could be sub-optimal, which will be explained in P5. **2)** Another reason why setting $L$ and $N$ would be better than $\alpha$ is that most of the related works require condition on $\alpha$ for the convergence statement to hold. However, we do not require such condition on $\alpha$, while the error can be controlled by $L$ and $N$. Moreover, the convergence statement holds for arbitrary $\alpha\in(0,1)$, $L$, and $N$.
>
> Lastly, we would like to emphasize that our main contributions are as providing the first analysis of reinforcement learning in a setting that closely resembles the deep Q-learning approach described in [3] within the TD-learning framework, and providing a rigorous analysis on the relation between the error bound and the mini-batch size ($L$), and the replay buffer size ($N$) within the TD-learning framework. We hope these contributions will provide valuable insights to the community. The related discussions have been newly included in the revised manuscript on page 12. We thank you again for your constructive feedback.
>
> **P2.** *Furthermore, the comment about Lakshminarayanan \& Szepesvari (2018) is not clear to me. Clearly that paper studies a more general setting with linear function approximation, and hence the step size requires knowledge of the problem structure, including the feature space.*
>
> **A2.** Thank you for the insightful comment. As the reviewer mentioned,  Lakshminarayanan \& Szepesvari (2018) studies a more general setup. However, in this previous paper, the step-size condition depends not only on the feature space but also on the minimum state-visitation distribution, $\mu^{\pi}_{\min}$, even in the tabular case. Therefore, we believe that the advantage is still preserved in a fairer setting. We have clarified this point on page 10 of the revised manuscript. Thank you for the attention on the matter.
>
>
> **P3.** *In this paper, rather than knowing the problem structure, we need the knowledge of the mixing time to set $N>\tau_{mix}$. How is it different than the prior work?*
>
> **A3.** In response to the comment, we would like to note that in most prior studies on the convergence of TD-learning with Markov observation model, finite-time bounds after $\tau^{\text{mix}}$ number of iterations is common in order to consider the mixing time. In this respect, we believe that the knowledge on $N>\tau^{mix}$ is fairly mild. We have clarified this in page 10 of the revised manuscript. Thank you for your attention on our manuscript.
>
>
> **P4.** *Similarly, in Theorem 3.6 you impose the strong assumption of $S_{-N}\sim \mu_S^{\pi}$. How is this assumption realistic, given that we do not know the stationary distribution of the underlying Markov chain?*
>
> **A4** The assumption is just for simplicity of the proof, and it was also used in the previous paper [1]. The assumption can be relaxed to the more practical case that the initial state distribution is arbitrary with some more efforts. To this end, we can bound the error caused by the initial distribution using the geometric mixing property of the uniformly ergodic Markov chain. Following the reviewer's comment, the related discussions have been newly added in the revised manuscript on page~11. Thank you for improving the clarity of our manuscript.

---

> > ### Author Response · Authors · 2024-05-27
> > **Reply to reviewer (Part 2)**
> >
> > **P5.** *In equation (3), you did not define what $e_{s_k}$ is. Overall, besides comparing the requirements on the step size, can you also discuss about the sample complexity of TD-learning with experience replay, and under similar conditions (and assumptions), can you compare its sample complexity with normal TD-learning? that way we can have a clear view on potential advantages of this algorithm.*
> >
> > **A5.** According to the reviewer's comment, the definition of $e_{s_k}$ has been newly added on page 5 of the revision.
> > As for the normal TD-learning, to the authors' knowledge, the known tight sample complexity result in terms of the mean-squared error is $\tilde{\mathcal{O}}\left(  \frac{t^{\text{mix}}}{(1-\gamma)^4 (\mu^{\pi}\_{\min})^4 } \frac{1}{\epsilon^2} \right)$  by [1]. Our sample complexity is $ \tilde{\mathcal{O}}\left(  \frac{\tau^{\text{mix}}|\mathcal{S}|^7|\mathcal{A}|^2 }{(1-\gamma)^6(\mu^{\pi}_{\min})^4} \frac{1}{\epsilon^4}\right)$, which is sub-optimal. Even though the sample-complexity is sub-optimal, as answered in P1, if the dataset is already given as in the offline RL setting, then we would require only small number of samples in addition, which covers different scenario compared to the normal TD-learning. Moreover, we can relax the condition on $\alpha$ for the convergence statement to hold while the error bound can be controlled by $L$ and $N$. In particular, the error bounds given in our paper are new in the sense that the it depends on the size of the replay buffer which can be controlled by adjusting the buffer size. For instance, the constant error ball caused by the constant step-size can become arbitrarily small by increasing the buffer size. Lastly, we want to emphasize that our main contribution is to first study the closest setting to the deep Q-network algorithm in [3] rather than proposing a new algorithm. The related discussions have been newly added in the revised manuscript on page 12. We thank the reviewer for the constructive comments.
> >
> > **References**
> >
> > [1] Jalaj Bhandari, Daniel Russo, and Raghav Singal. A finite time analysis of temporal difference learning with
> > linear function approximation. In Conference on learning theory, pp. 1691–1692. PMLR, 2018.
> >
> > [2] Agarwal, Rishabh, Dale Schuurmans, and Mohammad Norouzi. "An optimistic perspective on offline reinforcement learning." International Conference on Machine Learning. PMLR, 2020.
> >
> > [3] Mnih, Volodymyr, et al. "Human-level control through deep reinforcement learning." nature 518.7540 (2015): 529-533.

---

> > ### Comment · Reviewer_gLw3 · 2024-05-28
> > **Response**
> >
> > - The response to P1 is not satisfactory.
> > - Regarding A2 and A3, although you do not require the knowledge of \mu_min, but you need the knowledge of \tau^mix to set your N to be larger than the mixing time. These two requirements are in the same spirit. So I do not see any advantage here.
> > - Regarding A4, if you relax this assumption, are you going to need any extra assumption on the step size, N, or other parameters of the problem? Can you include that in the paper?

---

> ### Author Response · Authors · 2024-05-29
>
> **P6.** *The response to P1 is not satisfactory.*
>
> **A6.** : We apologize for the incomplete answer in A1. We provide a further detailed response in the following: First, we want to clarify that our contribution is to analyze the effect of using an experience replay buffer, which, despite its widespread use by practitioners, has not been discussed in theoretical detail. We have provided how we can view the role of $N$ and $L$, proving that the convergence rate depends on $L$ and $N$. We believe such analysis of widely used technique in practice could provide further insight into the community.
>
> Furthermore, our approach addresses a scenario that standard TD-learning analysis cannot: starting with a pre-existing dataset before transitioning to online learning, a topic of recent interest [6,7]. Our analysis shows that if the initial dataset is sufficiently large, i.e., with a large enough $N$, we can achieve a fast convergence rate by selecting a large $\alpha \in (0,1)$. In contrast, standard TD-learning analysis does not account for this situation.
>
> Lastly, the convergence statement is only mildly dependent on problem parameters. As answered in A7, we can set $\alpha \in (0,1)$, and $N$ being dependent on $\tau^{mix}$ is considerably less restrictive than requiring $\alpha \leqslant \mu^{\pi}_{\min}$.
>
> We have modified the discussion on page 11 and 12 in the revised manuscript. We thank you for your engagement in the discussion.
>
> **P7.** *Regarding A2 and A3, although you do not require the knowledge of $\mu_{min}$, but you need the knowledge of $\tau^{mix}$ to set your N to be larger than the mixing time. These two requirements are in the same spirit. So I do not see any advantage here*
>
> **A7.** Thank you for the valuable comment. Let us clarify the comparison with Lakshminarayanan \& Szepesvari (2018) and why $N$ being dependent on $\tau^{mix}$ is a mild condition: **1)** Lakshminarayanan \& Szepesvari (2018) only considered i.i.d. observation model whereas we cover Markovian observation model;  **2)** The mixing time, $\tau^{mix}$, is only logarithmically proportional to the minimum of the stationary distribution, i.e.,  $\tau^{mix}\approx \log \frac{1}{\bar{\mu}\_{\min}}$ where $\bar{\mu}\_{\min}=\min\\{   \min\_{s,s^{\prime}\in\mathcal{O}}\mu^{\pi}\_{S\_{\infty},S^{\prime}\_{\infty}}(s,s^{\prime}), \mu^{\pi}\_{\min}  \\}$. Assuming a uniform transition matrix, i.e., $\mathcal{P}(s,s^{\prime})=\frac{1}{|\mathcal{S}|}$ for all $s,s^{\prime}\in\mathcal{S}$, we have $\bar{\mu}^{\pi}\_{\min}=\frac{1}{|\mathcal{S}|^2}$, and we have $\tau^{mix}\approx 2\log(|\mathcal{S}|).$ We believe such logarithmic dependency on minimum stationary distribution is less stricter than the condition  $\alpha \leqslant \mu^{\pi}_{\min}=\frac{1}{|\mathcal{S}|}$, for example in Theorem 1 of  Lakshminarayanan \& Szepesvari (2018).
>
> Furthermore, we want to note that step-size or the time-horizon being dependent on mixing time is standard in the literature of analysis of Markovian observation model [1,2].
>
> We appreciate the engagement of the reviewer in the discussion and have incorporated it on page 9 of the revised manuscript.
>
> **P8.** *Regarding A4, if you relax this assumption, are you going to need any extra assumption on the step size, N, or other parameters of the problem? Can you include that in the paper?*
>
> **A8.**  We thank you for the constructive suggestion. The error of arbitrary initialization can be bounded by $O(\alpha)$ using the condition of geometric mixing property of the irreducible and aperiodic Markov chain in [1]. This will end up in the final error term $O(\alpha)$. However, such error is dominated by the $O(\alpha^{\frac{1}{2}})$ term in the error bound. Therefore, it will not significantly impact the convergence rate or the sample complexity. We have included the discussion on page 11 of the revised manuscript. Thank you for helping improving the clarity of the manuscript.
>
> **References**
>
> [4] Chen, Zaiwei, et al. "Finite-sample analysis of nonlinear stochastic approximation with applications in reinforcement learning." Automatica 146 (2022): 110623.
>
> [5] Qu, Guannan, and Adam Wierman. "Finite-time analysis of asynchronous stochastic approximation and $ Q $-learning." Conference on Learning Theory. PMLR, 2020.
>
> [6] Song, Yuda, et al. "Hybrid RL: Using both offline and online data can make RL efficient." The Eleventh International Conference on Learning Representations. 2022.
>
> [7] Ball, Philip J., et al. "Efficient online reinforcement learning with offline data." International Conference on Machine Learning. PMLR, 2023.

---

### Review · Reviewer_4S6k · 2024-05-18

**Summary Of Contributions:**

This paper looks at the finite-time analysis of the TD-learning when using an experience replay and where updates are done using minibatches drawn at random from the experience replay. It finds bounds on the noise induced by using minibatches, and uses it to analyze the convergence of the averaged iterate and the final iterate with respect to the true value function.

The paper also compares its results to other contemporary analyses.

**Audience:**

Yes

**Claims And Evidence:**

Yes

**Requested Changes:**

Ensure consistency and address issues above

**Strengths And Weaknesses:**

### Strengths:
* The paper takes on an ambitious but useful analysis
* From the proofs I have checked, the analysis seems to be correct.
* There are attempts to clarify and communicate the results to readers not immersed in such analysis and methods

### Minor Weaknesses:
* $k$ is overloaded to describe the Markov chain as well as the value function updates and the samples in the replay buffer. That makes it hard to keep track of which version of $k$ is being referred to sometimes.
* In Section 3.1, Replay Buffer $B_k^{\pi}$ has samples indexed as $O_{k-N+1}$, which implies $k$ is larger than $N$. But $k$ seems to refer to learning steps, which should start at $0$.
* $e_{s_k}$, while evident, doesn't seem to be defined
* Just before Section 3.3: the empirical estimation seems to be for the reward, not the return, and the reward is implied to only depend on $s$, but is defined to be over $(s, a, s')$ in Section 2. Even averaging over actions would imply that the reward here would be dependent on $(s, s')$

None of these nitpicks changes the analysis and the comparison with related methods though

---

> ### Author Response · Authors · 2024-05-27
> **Reply to reviewer**
>
> **P1.** *$k$ is overloaded to describe the Markov chain as well as the value function updates and the samples in the replay buffer. That makes it hard to keep track of which version of $k$ is being referred to sometimes.*
>
> **A1.** Thank you for the valuable feedback. The overload use of $k$ comes from the fact that we consider the online-learning setting. That is, at time-step $k$, $O_k$ is observed and $V_k$ is updated using a mini-batch, $M^{\pi}_k$, which is sampled from the replay buffer, $B^{\pi}_k$. To maintain clarity and consistency in our presentations, we have decided to retain the previous format. We hope the reviewer understands that this decision was necessary and unavoidable. Thank you for your understanding and consideration. Finally, we have clarified this in page 5 of the revised manuscript.
>
> **P2.** *In Section 3.1, Replay Buffer $B_k^{\pi}$ has samples indexed as $O_{k-N+1}$, which implies $k$ is larger than $N$. But $k$ seems to refer to learning steps, which should start at $0$.*
>
> **A2.** We greatly appreciate your valuable comment. The setting of our algorithm is as follows : For the first $N$ steps , only the data are collected and learning starts when $k=0$. We denote $O_{-N},O_{-N+1},\dots, O_{-1}$ as the first $N$ samples collected before learning starts. As learning starts at $k=0$, we observe $O_0$ and set $B^{\pi}_0=\{O_{-N+1},O_{-N+2},\dots, O_{0} \}$. For clarity, the discussions have been newly added on page 5 of the revised manuscript. Thank you for your attention to this matter.
>
>
> **P3.**  *$e_{s_k}$, while evident, doesn't seem to be defined*
>
> **A3.** Following the reviewer's comment, the definition has been newly added in the revision on page~5. Thank you for carefully examining our manuscript.
>
>
> **P4.** *Just before Section 3.3: the empirical estimation seems to be for the reward, not the return, and the reward is implied to only depend on $s$, but is defined to be over $(s, a, s')$ in Section 2. Even averaging over actions would imply that the reward here would be dependent on $(s, s')$*
>
> **A4.** We thank you for your valuable feedback. The estimate for the reward, which is introduced just before Section 3.3, is averaged over the action, $a$, and next state, $s^{\prime}$, not only for the action, $a$. We would like to note that the reward introduced in Section 2.2 depends on the state, action, and next reward, $s$, $a$, and $s^{\prime}$, respectively. The related discussions have been newly added on page 7 of the revised manuscript. Thank you for your attention on our manuscript.

---

### Author Response · Authors · 2024-05-27

We are grateful to the reviewers and the associate editor for their constructive comments on our manuscript, which are very helpful in improving the quality of the paper. In the following, we list the major changes we have made in response to these comments. Please note that the changes are marked with ${\color{blue}blue}$ fonts in the revised manuscript.

1) We have added detailed discussion on our contribution on revealing the relation between the convergence rate and the size of mini-batch and replay buffer, advantage of using a replay buffer, and comments on the sample complexity. (P2 and P5 of Reviewer gLw3, page 11 and 12 on the manuscript )
2) We have clarified the comparison with fitted Q iteration algorithm. (P2 of Reviewer EqyT, page 2 on the revised manuscript)

---

### Decision · Action_Editor_Jqhh · 2024-06-26

**Recommendation:** Accept as is

**Comment:**

While maybe not the end-all analysis of TD methods with replay buffers, it makes a distinct contribution to advance our theoretical understanding of such algorithms.  Theoretical evidence is sound, and the advance is of interest to TMLR's audience.  Straightforward acceptance.

**Audience:**

The reviewers also agree that the paper makes an original analysis with interesting, previously unproven, results.

**Claims And Evidence:**

The reviewers agree that the theoretical results and proofs are sound.